# TRAIN ONCE AND GENERALIZE:
# ZERO-SHOT QUANTUM STATE PREPARATION WITH RL

## ABSTRACT

Quantum state preparation forms an essential cornerstone of quantum information science and quantum algorithms. Designing efficient and scalable methods for approximate state preparation on near-term quantum devices remains a significant challenge, with worst-case hardness results compounding this difficulty. In this work, we propose a deep reinforcement learning framework for quantum state preparation, capable of immediate inference of arbitrary stabilizer states at a fixed system size post a training phase. Our approach scales substantially beyond previous works by leveraging a novel reward function. In our experiments on stabilizer states up to nine qubits, our trained agent successfully prepares nearly all previously unseen states, despite being trained on less than $10^{-3}\%$ of the state space – demonstrating significant generalization to novel states. Benchmarking shows our model produces stabilizer circuits with size 60% that of existing algorithms, setting a new state of the art in circuit efficiency. Furthermore, we show that this performance advantage is consistent across states with varying entanglement content. We also analyze the rate of increase of entanglement entropy across the prepared circuit, obtaining insight into the quantum entanglement dynamics generated by our trained agent. Finally, we prove our agent generalizes to (almost) the entire space of stabilizer states.

## 1 INTRODUCTION

At the heart of quantum information processing are quantum bits or qubits that can exist in arbitrary superpositions owing to the *coherence* properties of a quantum device. An increase in the number of qubits leads to an exponential increase in the complexity of the many-body state: preparing a general state of $n$ qubits efficiently on a quantum processor (called quantum state preparation, or QSP) remains a daunting task. The precise problem is as follows: given access to a target state $|\psi\rangle$, a set of allowed gate operations, and restrictions on qubit connectivity, can we come up with an (efficient) algorithm for a circuit-level construction of the state? The problem is of fundamental importance, being an essential primitive in the majority of modern quantum algorithms. QSP plays a major role in the Harrow–Hassidim–Lloyd (HHL) algorithm (Harrow et al., 2009) for solving linear systems, where a state preparation procedure is used to prepare quantum state $\sum b_i |i\rangle$ from classical unit vector $\boldsymbol{b}$. HHL in turn underpins many quantum machine learning (QML) algorithms (Biamonte et al., 2017; Liu et al., 2021). Quantum error correction (QEC), an essential ingredient in the realization of large-scale fault-tolerant quantum computers (Preskill, 2018), requires the efficient state preparation of logical code states (Gottesman, 2009). Apart from these, QSP also finds application in studying phase transitions and the ground state physics of many-body Hamiltonians (Lin & Tong, 2020; Dong et al., 2022).

A key challenge in the current landscape of quantum technology is the development of efficient state preparation methods suitable for noisy intermediate-scale quantum (NISQ) devices (Preskill, 2018). Many quantum algorithms that claim speedups in terms of oracle complexity rely heavily on state preparation through oracle calls (Aaronson, 2015). To achieve a practical quantum advantage, particularly in the NISQ era, efficient implementation of these oracles is crucial. Furthermore, due to limited coherence times and gate inaccuracies, current quantum hardware can only support a few thousand quantum gates (Preskill, 2018). As such, despite the theoretical worst-case hardness results, it remains critical to identify and develop practical state preparation protocols.

In recent years, deep reinforcement learning (DRL) has emerged as a powerful tool for solving search problems in complex state spaces. It has shown promise in solving general design problems, e.g. for combinatorial optimization (Dai et al., 2018), chip design (Mirhoseini et al., 2021) and even theorem proving (Wu et al., 2021). It is straightforward to model state preparation as a sequential prediction problem. An agent incrementally pieces together a circuit, adding an allowed quantum gate at each step, until the output of the circuit is (close to) the state of interest. The quantum system typically starts from a fiducial state $|\psi_0\rangle$. This brings us to our central line of inquiry.

*Can deep reinforcement learning offer a scalable and efficient solution to QSP?*

There has been much work on using DRL for state preparation. However, scaling to many qubits generally poses a challenge for current approaches owing to an exponentially increasing search space of possible circuits (Schneider et al., 2023). For this reason, much previous work is limited to states with a few qubits, or to states that are known to be realizable with a circuit of small size (He et al., 2021; Gabor et al., 2022; Wu et al., 2023; Kolle et al., 2024). A different, but arguably more critical problem is that many existing approaches (Schneider et al., 2023; Zen et al., 2024) require re-training for each choice of target state, which makes them usable only for the discovery of more efficient circuits for particular states of interest, not as a primitive that can replace an existing heuristic for preparing arbitrary states. An agent that does not need re-training to prepare unseen states will be called *zero-shot* in this work, to emphasize successful generalization to states not seen during training.

Taking a step to address these challenges, in this work we develop a reinforcement learning-based method to prepare *arbitrary* stabilizer states at a specified system size, gate-set and qubit connectivity. By focusing on the rich subset of stabilizer states, we are able to scale our method to the preparation of systems of up to 9 qubits. Our method lends itself to zero-shot agents: the training phase only needs to happen once for a given connectivity graph and gate set. Post the training phase, an arbitrary $n$-qubit state $|\psi\rangle$ belonging to the class of interest may be prepared just by providing the agent a classical description of the target state $|\psi\rangle$. To achieve this scaling in a sample-efficient manner, we motivate and analyze the novel class of moving-goalpost reward (MGR) functions.

Another important contribution of this work is the style of benchmarking state preparation agents. Apart from measuring circuit sizes of the output circuits and preparing states used in error-correcting codes, we examine the effect of entanglement on the produced circuits. We use brickwork circuits (Fig. 3(a)) to generate states with varying entanglement content and test the performance of the agent. Further, we analyze the entanglement dynamics of the agent *during* circuit preparation, leading to insights about the speed of preparation and redundancy in the produced circuits. The third important contribution is that of provable generalization: we show that our agents generalize to at least 95% of the state space, despite being trained on less than $10^{-13}$-$10^{-3}$% of the state space.

The paper is organized as follows. In Sec. 2, we first provide a short introduction on relevant aspects of quantum computation and reinforcement learning. After a discussion of previous work in Sec. 3, we move on to describe our proposal and novel reward function in detail in Sec. 4. Finally, the various experiments in Sec. 5 provide a deeper analysis of the performance of the trained agents.

## 2 BACKGROUND

### 2.1 QUANTUM COMPUTATION AND STABILIZER CIRCUITS

The state of a single qubit is described by a unit vector $|\psi\rangle$ in its Hilbert space $\mathcal{H} \cong \mathbb{C}^2$. We write $|\psi\rangle = a|0\rangle + b|1\rangle$, where $a, b \in \mathbb{C}$ and $\{|0\rangle, |1\rangle\}$ is a fixed orthonormal basis for $\mathcal{H}$. A general $n$-qubit state is a linear combination of the $2^n$ basis states $|z\rangle = \otimes_{i=1}^n |z_i\rangle \in \mathcal{H}^{\otimes n}$ with $z_i \in \{0, 1\}$. The *fidelity* between two quantum states is $\mathcal{F}(\psi, \phi) := |\langle\psi|\phi\rangle|^2$. A state is called entangled if it cannot be written as a tensor product of single-qubit states, for instance the Bell state $(|00\rangle + |11\rangle)/\sqrt{2}$. We quantify the entanglement content of a pure state $|\psi\rangle$ through the bipartite entanglement entropy: given any bi-partition $A \cup B$, we define $S(|\psi\rangle_{AB}) := -\text{tr}(\rho_A \log \rho_A)$, with $\rho_A := \text{Tr}_B(|\psi\rangle_{AB} \langle\psi|_{AB})$ where $\text{Tr}_B$ denotes the partial trace over subsystem $B$. In this work, we restrict to the half-chain entanglement entropy by choosing the bipartition $A = \{1 \leq i \leq n/2\}$. We point the reader to Nielsen & Chuang (2010) and App. A for more details on the theory of quantum circuits.

Stabilizer states are a restricted yet important class of quantum states described in a group-theoretic fashion as the common +1-eigenspace of an Abelian sub-group of Pauli Operators, rendering them classically simulable (Aaronson & Gottesman, 2004). They admit an equivalent representation as states that can be reached from the all-zeros state $|\mathbf{0}\rangle$ using *Clifford* circuits, i.e. unitaries that are a combination of $H$, $S$ and CNOT gates. Stabilizer states have immense use in the exploration of quantum information (Webb, 2016; Huang et al., 2020) and are also crucial for quantum error correction (QEC) (Gottesman, 1997; Nielsen & Chuang, 2010; Campbell et al., 2017; Ryan-Anderson et al., 2021). They are also applied beyond to measurement-based quantum computing (Raussendorf & Briegel, 2001; Patil & Guha, 2023), quantum-classical hybrid algorithms (Cheng et al., 2022; Ravi et al., 2022) and many-body physics (Sun et al., 2024).

Despite classical simulability, preparing stabilizer states optimally remains a challenge. It is known (Aaronson & Gottesman, 2004) that any stabilizer state can be prepared using $\mathcal{O}\left(n^2/\log n\right)$ gates, and that this bound is asymptotically tight. Quadratic circuit size coupled with the fact that the number of Clifford states grows as $2^{\mathcal{O}(n^2)}$ makes the search for optimal circuits difficult. Known optimal circuits have been limited to 6 qubits (Bravyi et al., 2022). Further, the (anti-)commutation and self-inverse properties of Clifford gates make it harder to reason about locally greedy search steps.

## 2.2 Reinforcement learning

In the Reinforcement Learning (RL) setting, an agent learns through interactions with an environment to maximize its *cumulative reward* across the interactions (Sutton & Barto, 2018). For a more complete introduction to RL, we refer the reader to App. A.3 and Sutton & Barto (2018). The environment is typically modeled as a Markov decision process, and the policy of the agent is modeled as a function $\pi : \mathcal{S} \times \mathcal{A} \to [0, 1]$ with $\pi(a|s)$ being the probability that the agent will take action $a$ when in state $s$. A policy $\pi$ along with a distribution $\mu$ over possible start states $s_0$ induces a distribution over traces $\tau = (s_0, a_0, s_1, \cdots, a_{T-1}, s_T)$ via $s_0 \sim \mu$, $a_i \sim \pi(\cdot|s_i)$, $s_{i+1} \sim p(\cdot|s_i, a_i)$ for each $i$. A reward function $r : \mathcal{S} \times \mathcal{A} \times \mathcal{S} \to \mathbb{R}$ defines the metric we want to optimize. Taking action $a$ while in state $s$ and landing up in state $s'$ yields a reward $r(s, a, s')$. The goal in RL is to find a policy $\pi^*$ that maximizes the expected cumulative reward or return

$$J(\pi) := \mathbb{E}_{\tau \sim (\mu, \pi)}\left[G(\tau)\right] = \mathbb{E}_{\tau \sim (\mu, \pi)}\left[\sum_{i=0}^{T-1} \gamma^i r(s_i, a_i, s_{i+1})\right]. \tag{1}$$

Here, $\gamma \in [0, 1]$ is the discount factor, describing the value of future actions in the present.

In the typical state preparation setting (and in our work), the state space is the set of states that we wish to prepare. The action space consists of allowed quantum gates. Taking an action $U$ corresponds to applying $U$ to the current state $|\psi\rangle$, with the new state after the action being $U|\psi\rangle$. The terminal state is usually the state we desire to prepare. However, in our work, we invert the preparation process and thus set the default state $|\mathbf{0}\rangle$ to be the lone terminal state instead. Throughout our experiments, we use the Proximal Policy Optimization (PPO) algorithm (Schulman et al., 2017), from the family of actor-critic methods, to train our agent. A detailed overview of the algorithm is provided in App. A.3.

## 3 Related work

There is a rich body of literature devoted to preparing quantum states. Classic methods include the Solovay-Kitaev construction (Dawson & Nielsen, 2005), the quantum Shannon decomposition (Shende et al., 2004), and more recently an inverse-free Solovay-Kitaev style construction (Bouland & Giurgica-Tiron, 2021). In recent times, many machine learning and deep reinforcement learning approaches have been examined (Kolle et al., 2024; Gabor et al., 2022; He et al., 2021; Zhang et al., 2019). Some DRL approaches tackle the related problem of quantum compiling, where one needs to prepare a (short) circuit describing a unitary (Fösel et al., 2021; Patel et al., 2024; Chen et al., 2022).

Owing to their diverse applicability, it is of great interest to find efficient circuits preparing stabilizer states. Algorithmic heuristics for the preparation of stabilizer states have been studied in great detail

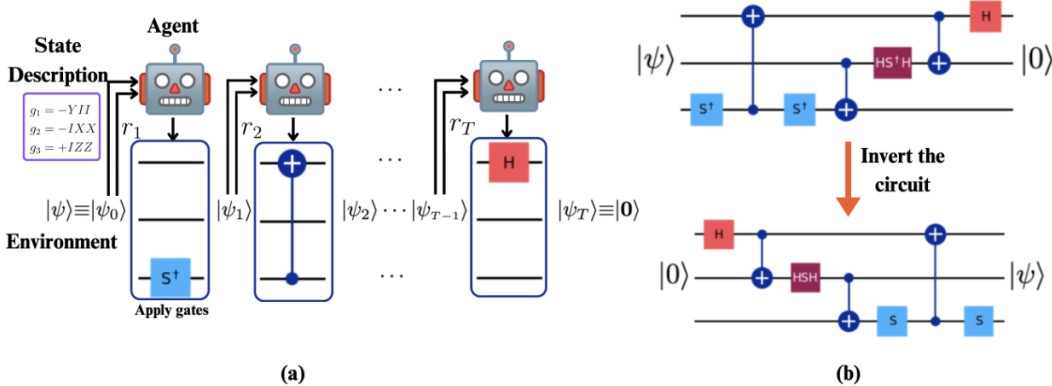

Figure 1: The proposed reinforcement learning framework. **(a)** The state to prepare is set as the *start* state, and the goal is to reach the all-zeroes state $|\mathbf{0}\rangle$. At each step, the agent interacts with $|\psi_t\rangle$ and proposes the next gate to apply. **(b)** After reaching $|\mathbf{0}\rangle$, we invert the circuit to prepare the target $|\psi\rangle$.

(Aaronson & Gottesman, 2004; Bravyi et al., 2021; 2022). The application of deep reinforcement learning to stabilizer QSP has been limited to the preparation of specific stabilizer states, typically centered around quantum error-correcting codes (Su et al., 2023; Zen et al., 2024). Approaches based on representing preparation as an optimization problem and using a SAT solver have also been studied (Peham et al., 2023; Schneider et al., 2023). Some recent works manage to address zero-shot inference using a reverse-preparation trick Wu et al. (2023); Zhang et al. (2020); Kremer et al. (2024); Wang & Wang (2024). Particularly, Huang et al. (2024a) uses a method using local circuit inversions to learn shallow unitaries.

## 4 ZERO-SHOT QUANTUM STATE PREPARATION WITH RL

**Problem 1** ((Approximate) Quantum State Preparation). *Given $n \in \mathbb{N}$, collection of $n$-qubit states $\mathcal{S} \subseteq \mathcal{C}^{2^n}$, starting state $|\psi_0\rangle$, a set of gates $\mathcal{A}$ induced by a collection $\mathcal{G}$ of allowed gates and qubit connectivity graph $\mathcal{N}$, and $\epsilon > 0$, find an (efficient) algorithm that upon input any state $|\psi\rangle \in \mathcal{S}$ returns a (short) circuit $\mathcal{C}$ such that $\mathcal{F}(|\psi\rangle, \mathcal{C}|\psi_0\rangle) \geq 1 - \epsilon$.*

Here, the qubit connectivity graph (also called coupling map) $\mathcal{N} = (V, E)$ with $V = \{q_i\}_{i=1}^n$ being the set of qubits and edge $e = \{q_i, q_j\}$ representing the fact that two-qubit gates may be applied to the joint system of $q_i$ and $q_j$. A set of allowed single and two-qubit gates $\mathcal{G}$ induces a collection of allowed $n$-qubit gates $\mathcal{A}$ with each single-qubit gate $g \in \mathbb{C}^{2\times 2}$ contributing $|V| = n$ gates – apply $g$ to the $i^{\text{th}}$ qubit, leave the rest as is – and each two-qubit gate $g \in \mathbb{C}^{4\times 4}$ contributing $|E|$ gates – apply $g$ to the joint system of $q_i$ and $q_j$ for each edge $\{q_i, q_j\}$ – to $\mathcal{A}$. For example, at $n = 3$ with gate-set $\{H, S, \text{CNOT}\}$ and connectivity graph $\{\{1, 2\}, \{1, 3\}\}$, the induced collection of gates comprises $H \otimes I \otimes I, I \otimes H \otimes I, I \otimes I \otimes H, S \otimes I \otimes I, I \otimes S \otimes I, I \otimes I \otimes S, \text{CNOT}_{1,2} \otimes I_3, \text{CNOT}_{2,1} \otimes I_3, \text{CNOT}_{1,3} \otimes I_2$ and $\text{CNOT}_{3,1} \otimes I_2$. Here $\text{CNOT}_{i,j}$ is the CNOT gate applied to qubits $q_i$ and $q_j$ with $q_i$ the control and $q_j$ the target. $I_k$ is the identity gate applied to qubit $q_k$.

### 4.1 ENABLING ZERO-SHOT INFERENCE

A typical formulation of QSP in the RL paradigm is to let the environment start in $|\psi_0\rangle$ with the agent changing the state of the environment by applying a gate $a \in \mathcal{A}$ at each step till the state of the environment is $\epsilon$-close to the target $|\psi\rangle$. The trajectory of the agent would then yield the desired circuit $\mathcal{C}$. An intrinsic drawback of this formulation is that the algorithm would need to be trained for each $|\psi\rangle \in \mathcal{S}$ as the terminal states of the environment in an RL setting are fixed.

The main observation is that we can exploit the fact that the environment's start state can vary across the training according to a distribution $\mu$. In view of this, consider $|\psi\rangle$ being the *start* state of

the environment, drawn uniformly from $\mathcal{S}$, with the target being (all states which are $\epsilon$-close to) the fiducial $|\psi_0\rangle$. The agent picks actions from the inverted action space $\mathcal{A}^\dagger = \{a^{-1} : a \in \mathcal{A}\}$ and attempts to prepare $|\psi_0\rangle$. A successful trajectory leads to a circuit $\tilde{C}$ with $\mathcal{F}(|\psi_0\rangle, \tilde{C}|\psi\rangle) = \langle |\psi_0\rangle |\tilde{C}|\psi\rangle\rangle^2 \geq 1 - \epsilon$. Further notice that

$$\mathcal{F}(\tilde{C}^{-1}|\psi_0\rangle, |\psi\rangle) = \langle\psi_0|\tilde{C}|\psi\rangle^2 = \mathcal{F}(|\psi_0\rangle, \tilde{C}|\psi\rangle) \geq 1 - \epsilon,$$

implying that the circuit $C = \tilde{C}^{-1}$ satisfies the conditions of the QSP problem, i.e. takes $|\psi_0\rangle$ to within $\epsilon$-fidelity of $|\psi\rangle$. Further, $C$ can be computed easily from $\tilde{C} = a_k^{-1}a_{k-1}^{-1}\cdots a_1^{-1}$ as $C = a_1\cdots a_{k-1}a_k$, i.e., the sequence of inverted actions along the reversed trajectory of the agent. This immediately gives us the RL formulation, see Tab. 1.

With this inverse preparation procedure, the start state $|\psi\rangle$ changes each episode, while the target state $|\psi_0\rangle$ is constant. In particular, the agent is being forced to prepare $|\psi_0\rangle$ from any starting $|\psi\rangle$ – and hence any $|\psi\rangle$ from $|\psi_0\rangle$ – avoiding the need to re-train the agent to prepare different $|\psi\rangle \in \mathcal{S}$.

We demonstrate the performance and scaling behavior of our algorithm by preparing stabilizer states, i.e. $\mathcal{S} = \mathbb{S}_n$, the set of $n$-qubit stabilizer states. We set $\epsilon = 0$, i.e. we target *exact* stabilizer state preparation. A good policy can potentially obtain high rewards (corresponding to effective and efficient circuits) on unseen states, which is empirically observed in our experiments (Tab. 3) and proven using generalization bounds in Sec. 5. A good reward function is the main bottleneck to guiding the agent towards a good policy; we describe our novel reward function in the following section.

Table 1: The proposed RL framework for state preparation

| Component | Description |
|---|---|
| State Space $\mathcal{S}$ | The set of $n$-qubit quantum states to be prepared. |
| Action Space $\mathcal{A}$ | The *inverse* of all gates in the induced collection of $n$-qubit gates. |
| Transition Function $p(s, u, s')$ | Deterministic: $s' = u \cdot s$ if action $u \in \mathcal{A}$ is applied to state $s \in \mathcal{S}$. |
| Starting Distribution $\mu$ | Uniform over $\mathcal{S}$. |
| Terminal state | $|\psi_0\rangle \equiv |\mathbf{0}\rangle$. |

### 4.2 Moving-Goalpost Reward (MGR) functions

In this section, we first discuss issues with typical reward functions used for state preparation, and use an experimental insight to derive a novel reward function. We finish the section with details of the precise reward function used in our experiments.

In quantum state preparation, a typical choice of reward function is the fidelity to the target $\Phi(s) := \mathcal{F}(s, |\psi\rangle)$, that is, $r(s_i, a_i, s_{i+1}) = \Phi(s_{i+1})$. In our reverse-preparation, the last term is $\mathcal{F}(s_{i+1}, |\mathbf{0}\rangle)$. However, we find that this reward does not learn – the cumulative reward obtained from this reward function does not reflect maximum *final* fidelity, which is our true goal. The agent might, for example, learn to stay close to the target without actually terminating the episode, e.g. an agent in state $(|00\rangle + |11\rangle)/\sqrt{2}$ can apply a CNOT gate repeatedly, always staying at a fidelity of $1/2$ to target state $|00\rangle$; this is optimal for the agent. Indeed, we observe precisely this style of behavior in Fig. 2.

Another choice of reward is the incremental fidelity $r(s_i, a_i, s_{i+1}) = \gamma\Phi(s_{i+1}) - \Phi(s_i)$ (here $\gamma$ is the discount parameter). It has been shown to work well for preparing a particular state, but we find that it does not learn to prepare arbitrary states in our framework (see Fig. 2).

We attribute the failure of this reward to learn to two reasons. Firstly, a gate ($H, S, HSH$ or CNOT) applied to a state $|\psi_i\rangle$ is very likely not to increase the fidelity to $|\mathbf{0}\rangle$. We experimented with 1000 uniformly random 6-qubit states, applying every one of 48 gates induced from our gateset to each state. We found that 83.4% of the actions yielded no change in fidelity, 10.0% reduced fidelity and only 6.6% improved it. The fidelity to the target across the optimal preparation of many states is not monotonic and involves sections where the fidelity decreases; the incremental reward penalizes these steps, possibly discouraging the agent from re-trying the same actions. We try to address these shortcomings by rewarding an agent suitably for an increase in fidelity, but not penalizing the agent

as much for an equal/smaller fidelity. Notice that it is unwise to reward the agent at every step with an increase in fidelity – an optimal policy would simply be to drop fidelity, increase it, and repeat till the episode is truncated. Such policies are allowed by the self-inverse nature of the quantum gates used. Thus, we choose to reward the agent only when fidelity surpasses *all previous fidelities* seen so far. The MGR reward function follows naturally.

We now introduce a class of rewards that we call Moving-Goalpost Reward (MGR) functions. These reward functions repeatedly set a performance baseline, reward an agent for beating it, and update the baseline. The formal definition follows.

**Definition 1.** *A function $\Phi : \mathcal{S} \to [0, 1]$ function on the state space satisfying $\Phi(s_T) \geq \Phi(s)$ for all terminal states $s_T \in \mathcal{S}$ and arbitrary states $s \in \mathcal{S}$ is called a potential function on $\mathcal{S}$.*

A potential function is a heuristic whose value indicates the closeness of a state to being terminal. Note that we do not place any restriction on the convexity of $\Phi$. We now define the general class of MGR functions.

**Definition 2 (MGR function).** Let $\Phi$ be a potential function on $\mathcal{S}$. Reward function $r : \mathcal{S} \times \mathcal{A} \times \mathcal{S} \to \mathbb{R}$ is a $\Phi$-MGR reward function if for every $k \geq 0$ and $k$-step trajectory $\tau = (s_0, a_0, s_1, \cdots, s_k, a_k, s_{k+1})$, we have

$$r(s_k, a_k, s_{k+1}) = \begin{cases} f(\Phi(s_{k+1}), M_k) & \Phi(s_{k+1}) > M_k \\ g(\Phi(s_{k+1}), M_k) & \text{otherwise} \end{cases},$$

where $f, g : [0, 1]^2 \to \mathbb{R}$ satisfy $f(x, y) \geq g(x, y) \, \forall \, x, y$ and $M_k := \max_{0 \leq i \leq k} \Phi(s_i)$.

Consider the specific instantiation $f(\Phi(s_{k+1}), M_k) = \gamma \Phi(s_{k+1}) - M_k$ and $g(\Phi(s_{k+1}), M_k) = -(1 - \gamma)M_k$ and call the associated MGR function MGR-VANILLA. For this reward, it can be proved that maximizing return, i.e. discounted cumulative reward over a $T$-length episode is equivalent to maximizing $M_T = \max_{0 \leq i \leq T} \Phi(s_i)$, i.e. reaching a terminal state during (and hence at the end of) the episode. In particular, we show in App. B.1 using a telescoping argument that the discounted cumulative reward over trace $\tau = (s_0, a_0, \cdots, s_{T-1}, a_{T-1}, s_T)$ is

$$G(\tau) = \sum_{i=0}^{T-1} \gamma^i r(s_i, a_i, s_{i+1}) = \gamma^T M_T - \Phi(s_0),$$

so maximizing $G(\tau)$ corresponds to reaching a terminal state. A side effect is that reducing $T$ also increases $G(\tau)$, so short circuits are preferred. However, since $\gamma \approx 1$ in our experiments, we instead used a small negative constant at each step to indicate a preference for shorter circuits. In our experiments, we slightly modify the above MGR reward function to penalize non-increasing fidelity as little as possible; we call the reward MGR-OURS:

$$r(s_k, a_k, s_{k+1}) := \begin{cases} \gamma \Phi(s_{k+1}) - M_k - \alpha & \Phi(s_{k+1}) > M_k \\ -\alpha & \text{otherwise} \end{cases}.$$

We use the reward MGR-OURS in all our implementations as it was seen to reduce training time, see Fig. 2. To ensure the environment stays Markovian, we augment the state $s_k$ with $M_k$.

We take a moment here to contrast our approach to existing methods that use a similar reverse-preparation idea, in various ways. Our method achieves zero-shot inference using a reverse-preparation trick. However, the main driver is actually our novel reward function, which yields scalable and sample-efficient agents, while staying zero-shot. We are able to scale far beyond Wu et al. (2023); Zhang et al. (2020), which address the general state preparation problem on 1-2 qubits. In fact, we have already used the same framework and reward function to prepare agents that successfully solve the zero-shot general state preparation problem for up to three qubits with $\varepsilon = 0.95$. However, our focus here is on stabilizer states and the general setting requires more comprehensive investigation with suitable adaptations, which is beyond the current scope.

The parallel work (Kremer et al., 2024) scales stabilizer state preparation to 11 qubits on various architectures. Our approach, though scaling only up to 9 qubits, substantially improves upon this work in terms of sample efficiency: we use at most 10M-20M training steps, while Kremer et al. (2024) use over 1B steps.

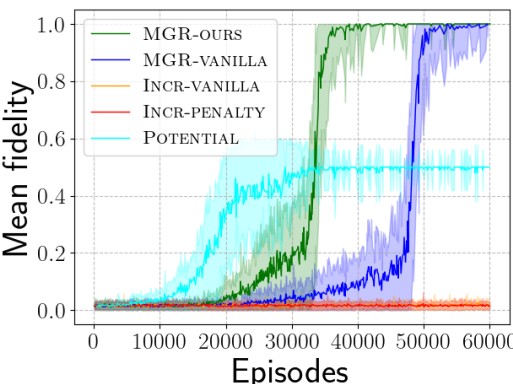

Figure 2: Analysis of different reward functions to the training of the 6-qubit fully connected agent. INCR refers to the incremental fidelity $\gamma\Phi(s_{i+1}) - \Phi(s_i)$. INCR-PENALTY has an extra $-\alpha$ term. $\alpha$ is chosen to be $1/T^*$, with $T^*$ being the maximum length of an episode. The POTENTIAL reward sets $r(s_i, a_i, s_{i+1}) = \Phi(s_{i+1})$. MGR-OURS converges faster than MGR-VANILLA.

## 5 EXPERIMENTAL RESULTS

To demonstrate the performance of our framework, we train the agent to prepare stabilizer states, both with unrestricted/full/all-to-all connectivity and linear/local connectivity, demonstrating state-of-the-art performance in both cases. We use the number of gates in the circuit as our main evaluation metric for circuit size; smaller is better. The agents are tested on increasingly entangled brick-work states (see Fig. 3(a)) to understand the dependence of the prepared circuits on the input state's entanglement content. We further contrast the entanglement dynamics generated by random stabilizer circuits with the dynamics generated by our trained agents. The probe for studying the entanglement dynamics is the canonical half-chain entanglement entropy (as defined in Sec. 2.1). We compare the performance of our RL model with two other methods (Aaronson & Gottesman, 2004; Bravyi et al., 2021) for arbitrary stabilizer state preparation. We note that both these methods use the Pauli gates $X$, $Y$, $Z$, $H$, $S$, controlled-NOT and SWAP gates with full connectivity for state preparation. All circuits prepared by the agent have a fidelity of 1.0.

**State and Action spaces**. For each $n$ and both connectivities, we use the set of $n$-qubit stabilizer states as our state space. Each stabilizer state is represented in flattened tableau form (Aaronson & Gottesman, 2004), only including its stabilizers, so that each state is represented by a $(2n^2 + n)$-dimensional bit-vector. For the action space, we use different allowed gatesets for each connectivity. Both gate-sets are realistic, for example in trapped-ion-based quantum computers (Cirac & Zoller, 1995), a promising candidate for quantum computation. For the fully connected agents, our gateset $G$ consists of $H$ (Hadamard), $S$ (Phase), CNOT and $HSH$ (conjugated phase) gates. The inclusion of the conjugated phase gate $HSH$ ensures symmetry within the gate set because it provides an operation for the $X$ component that mirrors the effect of $S$ on the $Z$ component. While $S$ modifies the $Z$ component of the tableau, the $HSH$ gate equivalently modifies the $X$ component. This symmetry is useful since the tableau is also symmetric in $X$ and $Z$. It is not unfair to treat $HSH$ as a single gate; it is as easy as $S$ to apply (Evered et al., 2023) – note that $HSH$ is simply a $\pi/2$-rotation about the $x$-axis just as $S$ is a $\pi/2$-rotation about the $z$-axis. For the local case, we use the $H$, $S$, $X$, $Y$, $Z$ and CNOT gates. Despite using a more restrictive gate-set for our agents compared to baselines, our methods achieve substantially shorter circuits (see Tab. 3).

**Testbench details**. We use two testbeds. One consisting of uniformly random stabilizer circuits and the other comprising (roughly) uniformly sampled *brickwork circuits* (see Fig. 3(a)) of different depths. The random stabilizer circuits were sampled using the Stim API (Gidney, 2021). Each brick-work circuit of depth $d$ was constructed by choosing each "brick" to be an independently sampled random 2-qubit stabilizer circuit. The first test-bench serves to estimate the average performance of the agent; the second examines the entanglement dynamics of the induced preparation algorithm of the agent. Finally, we also additionally sampled $2,000$ independent uniformly random stabilizer circuits for each $n = 5, 6, 7, 9$ to provide the empirical data to prove our generalization bounds.

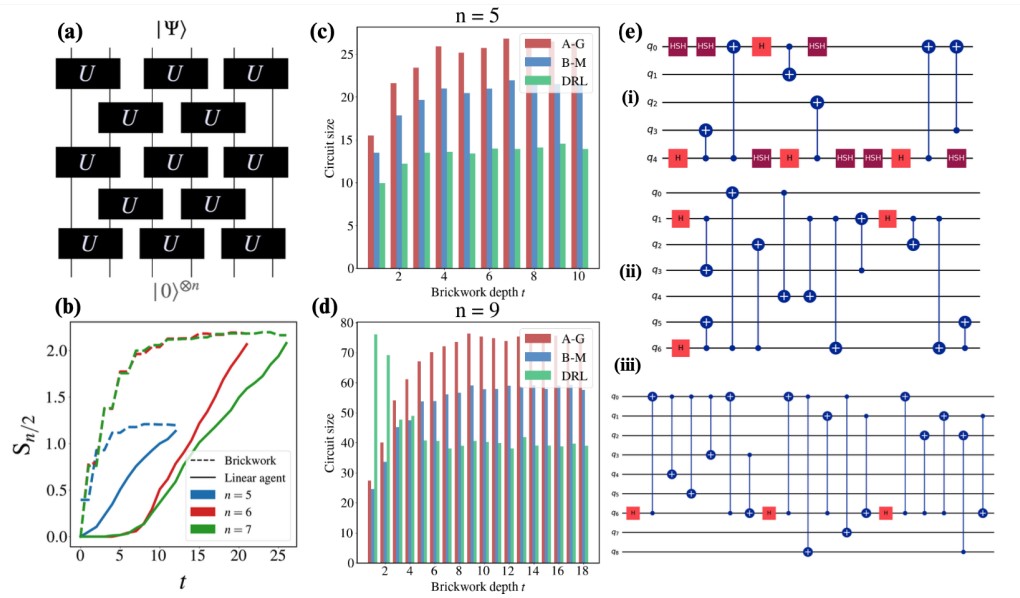

Figure 3: **(a)** A schematic of brickwork Clifford circuits, where each $U$ is sampled independently from the 2-qubit Clifford group. **(b)** The progression of entanglement entropy during the preparation of volume-law entangled $2n$-depth brickwork states (solid) vs the entanglement entropy of $n$-qubit brickwork states of increasing depth $t$ (dashed) **(c-d)** Benchmarking our model on increasingly deep brickwork circuits for (c) $n = 5$ and (d) $n = 9$ qubits. A-G and B-M refer to the stabilizer preparation methods in Aaronson & Gottesman (2004) and Bravyi et al. (2021) respectively, and DRL refers to our framework (full connectivity). **(e)** A zero-shot preparation of the logical $|0\rangle_L$ states of three popular codes with full connectivity.

**Implementation details**. To facilitate quick learning, especially at the start of training, we artificially terminate episodes after a fixed maximum time-step $T^*$. The pairs $(n, T^*)$ used in our experiments are $(5, 50)$, $(6, 80)$, $(7, 80)$ and $(9, 127)$. The discount factor is fixed at $0.99$ for $n < 9$ and $0.9$ for $n = 9$. Training hyper-parameters can be found in App. D.2. We implement a version of PPO based on Morales (2020) in `PyTorch` and simulate stabilizer states using Stim (Gidney, 2021). The environment is vectorized for parallel training on a single GPU. The agent is allowed five attempts at preparing each state; we pick the best one (shortest circuit size). This is done mostly to give the agent the chance to discover even shorter circuits; almost every attempt yields a successful preparation nonetheless. Finally, for the local agents, we augment MGR with the incremental Jaccard distance between $|\mathbf{0}\rangle$ and the current state since we found that this improved sample efficiency.

**Computational costs of training**. Our agent is sample-efficient and accordingly, training times are short: Training for $40,000$ episodes with $n = 5$ takes 15 minutes on a single NVIDIA A100 GPU. At $n = 7$, training takes 3 hours (130k episodes) and at $n = 9$ takes 5 hours (180k episodes).

**Preparing stabilizer states, full connectivity**. In this set of experiments, we train four agents to prepare arbitrary stabilizer states with number of qubits $n = 5, 6, 7, 9$ with full connectivity. Our proposed RL method performs substantially better – circuit sizes are $60\%$ that of baselines – than other methods at most brickwork sizes, especially at high entanglement. Tab. 3 shows the results of preparing 200 randomly sampled stabilizer states of each size. We also provide an analysis of the CNOT gate counts of circuits prepared by our trained agents in App. D.1. We observe efficient use of these gates despite offering no extra bias towards minimizing the usage of two-qubit gates.

Figs. 3(c) and 3(d) show the comparison of circuit size with existing approaches for stabilizer states for $n = 5$ and $n = 9$ respectively. We benchmark each method with $N = 100$ random brickwork circuits (see Fig. 3(a)) for each depth $t \in \{1, 2, \cdots, 2n\}$. Brickwork circuits were chosen to explore the performance of the RL method in preparing highly entangled states.

Table 2: Circuit size (↓) comparison with baselines, averaged across 200 uniformly random stabilizer states of the appropriate size.

| Algorithm | 5-qubit | 6-qubit | 7-qubit | 9-qubit |
|---|---|---|---|---|
| Aaronson & Gottesman (2004) | $26.00 \pm 6.37$ | $36.43 \pm 7.25$ | $48.13 \pm 7.29$ | $76.56 \pm 8.23$ |
| Bravyi et al. (2021) | $21.10 \pm 4.88$ | $29.77 \pm 5.97$ | $38.50 \pm 5.96$ | $59.24 \pm 7.57$ |
| RL (linear connectivity) | $\mathbf{15.52} \pm 3.25$ | $\mathbf{21.68} \pm 3.32$ | $\mathbf{30.18} \pm 4.09$ | - |
| RL (full connectivity) | $\mathbf{12.83} \pm 2.40$ | $\mathbf{17.86} \pm 2.88$ | $\mathbf{24.36} \pm 3.47$ | $\mathbf{41.92} \pm 5.91$ |

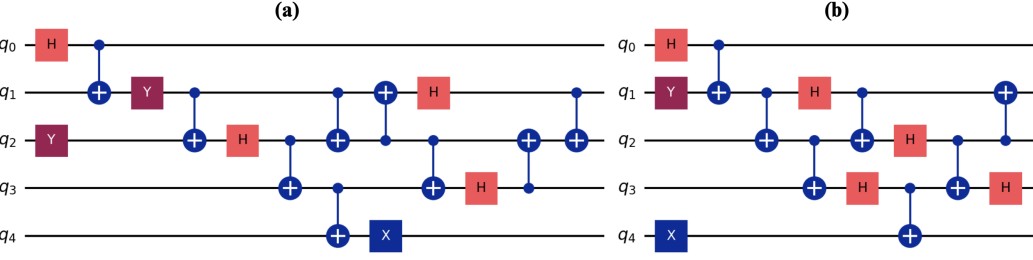

Figure 4: Preparing logical states $|0\rangle_L$ **(a)** and $|1\rangle_L$ **(b)** of the $[5, 1, 3]$ perfect code (Laflamme et al., 1996) when constrained to a linear connectivity.

**Stabilizer QSP with linear connectivity**. In this set of experiments, we construct circuits for stabilizer states with $n = 5, 6, 7$ with restricted connectivity: the connectivity graph $\mathcal{N}$ is a line, i.e. it only contains edges ($\{q_i, q_{i+1}\}$ for $i \in \{1, \cdots, n-1\}$). We chose this connectivity since it is often a subgraph of the connectivity graph of real quantum architectures, and so circuits generated with this connectivity may be used on these architectures directly. The gateset uses the single-qubit Pauli gates $X$, $Y$ and $Z$ in addition to the Clifford gate-set $H$, $S$ and CNOT. The results are shown in Tab. 3; we note that even restricting to local gates, the RL approach generates around $30\%$ shorter circuits.

**Preparation of some typical states used in QEC**. Fig. 3(e) shows the RL agent's attempt at preparing the logical state $|0\rangle_L$ for the (i) $[5, 1, 3]$ perfect code (Laflamme et al., 1996), (ii) $[7, 1, 3]$ Steane code (Steane, 1996) and the (iii) $[9, 1, 3]$ Shor code (Shor, 1995). Additionally, Fig. 4 shows the circuits prepared by the agent for the logical states of the perfect code when restricted to linear connectivity. Note that the agent was never explicitly trained to prepare any of these states.

We remark here one insight about the preparation of the logical zero state for the Shor code. The state itself is a tensor product of three copies of the 3-qubit GHZ state; we note that our agent is indeed simply preparing the GHZ gate three times, one after the other. This in turn signals that the efficiency of such an agent is intimately linked with whether the correlations in the quantum state are being captured by the algorithm. We further comment on this below.

**Entanglement dynamics**. This analysis concerns $n$-qubit states generated from brickwork circuits of depth $2n$ and the agent restricted to a linear connectivity, both of which generate *local* dynamics with qubit interactions on $\mathcal{O}(1)$ number of qubits at each step. Given a circuit $C = [U_1, \cdots, U_k]$ prepared by the local agent for a brickwork state, we compute the entanglement entropy $S(t)$ of the intermediate states $|\psi_t\rangle = U_t U_{t-1} \cdots U_1 |\mathbf{0}\rangle$ for each $1 \leq t \leq k$ as $S(t) = S_{N/2}(|\psi_t\rangle)$. The goal of this experiment is as follows: for preparing highly entangled $n$ qubit states with *local* dynamics, there are known bounds on the rate at which information can spread (Chen et al., 2023) owing to the locality of the dynamics. From the point of view of circuit optimization, this implies that the rate of correlations generated by the agent is important: a strong suppression of the entanglement rate $dS(t)/dt$ would imply longer circuits generated by the agent. Random brick-work circuits on $n$ qubits are prototypical examples of local quantum dynamics, displaying an initial linear increase in $S(t)$ followed by saturation at $t \lesssim 2n$. It is thus of interest to contrast the entanglement dynamics generated by our local agents with the random brickworks.

The dashed lines in Fig. 3(b) denote the entanglement entropy dynamics of $n$-qubit brickworks averaged over 1000 realizations, and the solid lines show $S(t)$ averaged for preparing over 200 volume-law entangled states. The entanglement entropy of the states prepared corresponds to the saturation value of the dashed lines. Upon close inspection, one bottleneck of the algorithm is seen to be the initial 'exploration' phase where $S(t)$ does not increase, after which the agent generates linearly increasing entropy although with a rate lesser than the brickwork. This is the underlying information-theoretic interpretation of the efficiency, whereby scaling this algorithm further crucially depends on the scaling of the exploration phase as well as the post-exploration rate as a function of $n$. Moreover, the intermediate entanglement entropies for a particular state are generally observed to monotonically increase (not shown), suggesting low redundancy in the use of the entangling CNOT gates on the part of the agent and further justifying our benchmarks.

**Theoretical analysis of agent generalization**. All experiments so far involve the agent preparing arbitrary states not seen during training, indicating a generalization to unseen states. We provide rigorous justification for this observation, showing that with probability at least $1 - 10^{-10} \approx 1$, the agent generalizes to at least 95% of the state space. In particular, we show in App. C that the following concentration result holds.

**Proposition 3 (Informal).** *F*ix $\varepsilon, \delta > 0$. Let $\mathcal{A}$ be a state preparation agent and $X$ to be the random variable over the uniform distribution on $\mathcal{S}_n$ with $X(|\psi\rangle) = 1$ whenever $\mathcal{A}$ successfully prepares $|\psi\rangle$ and 0 otherwise. Let $\bar{X}$ be the average value of $X$ across $N$ uniformly sampled states $|\psi\rangle$ and suppose that $N \geq \frac{1}{2\varepsilon^2} \log \frac{1}{\delta}$. Then with probability at least $1 - \delta$ over the choice of samples $(X_1, \cdots, X_N)$,

$$\mathbb{E}[X] \geq \bar{X} - \varepsilon.$$

We set $\delta = 10^{-10}$, $\epsilon = 0.05$ and $N = 2000$. For each $n = 5, 6, 7, 9$, we sample $N$ uniform states from $\mathcal{S}_n$ and run our algorithm. We find that our algorithm prepares all of them exactly, implying $\bar{X} = 1$. It follows that $\mathbb{E}[X] \geq 0.95$ for each $n$, which means that at least 95% of $\mathcal{S}_n$ *will* be prepared successfully by the agent. This implies massive generalization: $|\mathcal{S}_n|$ is 2.4M, 315M, 81.3B and $4.38 \times 10^{16}$ for $n = 5, 6, 7$ and 9 respectively. Our result says that our agent will successfully prepare at least 2.3M, 300M, 77.2B and $4.07 \times 10^{16}$ states – which are many many orders larger than the 10-20M states seen during training. This large-scale generalization provides insight into the empirical success of the agent on virtually all states.

## 6 CONCLUSIONS AND FURTHER WORK

In this work, we have demonstrated that deep reinforcement learning can facilitate immediate inference on arbitrary stabilizer states without needing re-training. This is achieved through a highly sample-efficient reverse preparation approach utilizing a novel reward function. Our experiments perform a thorough analysis of the agent's inference capabilities and the potential of this approach to improve performance over existing baselines. We show that our agents remain efficient across the spectrum of entanglement content of the target. Additionally, we provide information-theoretic insights into the dynamics and (low) redundancy of the trained agents and also present compelling arguments for their generalization capabilities. An important contribution is that our algorithm takes time proportional to the size of the circuit returned, which is of the same order as traditional rule-based approaches to state preparation. This suggests its use as a replacement for these algorithms, at least for a small number of qubits. This provides promise for the integration of RL-based methods into real quantum computing environments for transpilation.

An important limitation is that our algorithm does not scale yet to qubit count in the hundreds or thousands, which traditional approaches can handle, even if sub-optimally. This brings us to the first direction of future inquiry: an imminent task is designing *local* RL algorithms which operate on an $\mathcal{O}(1)$ number of qubits by making local queries to the target state (Huang et al., 2024b;a). Furthermore, while the current work addresses the question of finding optimal circuits to be implemented in a lab setting, it is indeed possible to restrict the agent's knowledge of the state under preparation to only local Pauli measurements. Moreover, understanding the fundamental bounds on the scaling of the exploration phase as the number of qubits in the target state is an important task to gain insight into the fundamental limits of such algorithms. Overall, our findings illustrate further the promise of employing deep RL methods for efficient state preparation on near-term quantum systems.

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

# A ADDITIONAL BACKGROUND

## A.1 QUANTUM COMPUTATION AND QUANTUM CIRCUITS

We briefly introduce the principles of quantum computing, quantum circuits and stabilizer states. For more elaborate discussions of these topics we recommend Nielsen & Chuang (2010); Aaronson & Gottesman (2004); Yoder (2012).

The state of a single qubit is described by a unit vector $|\psi\rangle = a |0\rangle + b |1\rangle$, where $a, b \in \mathbb{C}$ with $|a|^2 + |b|^2 = 1$ and $\{|0\rangle, |1\rangle\}$ is a fixed orthonormal basis, often called the computational basis, spanning the single qubit Hilbert space $\mathcal{H} \cong \mathbb{C}^2$. We use the Dirac notation here, where $|\psi\rangle = (a, b)^T$ is a column vector and $\langle \psi| = |\psi\rangle^\dagger$ the dual row vector. The state of a bipartite quantum system which consists of two parts $A$ and $B$ lives in the tensor product space of the individual Hilbert spaces $\mathcal{H}_A$ and $\mathcal{H}_B$, $|\psi\rangle_{AB} \in \mathcal{H}_A \otimes \mathcal{H}_B$. It follows that a general $n$-qubit state is a linear combination of the $2^n$ basis states $|z\rangle = \otimes_{i=1}^n |z_i\rangle \in \mathcal{H}^{\otimes n}$ ($z_i \in \{0, 1\}$). The *fidelity* between two quantum states is $\mathcal{F}(\psi, \phi) := |\langle \psi | \phi \rangle|^2$. An $n$-qubit state is called entangled if it cannot be written as a tensor product of single-qubit states. For instance, $(|00\rangle + |11\rangle)/\sqrt{2}$ is an entangled state.

A common method to quantify the entanglement content of a pure state $|\psi\rangle$ is through the bipartite entanglement entropy: given any bipartition $A \cup B$ of the qubits, we define $S(|\psi\rangle_{AB}) := -\text{tr}(\rho_A \log \rho_A)$, with $\rho_A := Tr_B(|\psi\rangle_{AB} \langle \psi|_{AB})$ where $Tr_B$ denotes the partial trace over subsystem $B$. While working with a chain of $n$ qubits in this work, we restrict to the half-chain entanglement entropy by choosing the bipartition $A = \{1 \le i \le n/2\}$.

A quantum gate or operation on a system of qubits is a unitary ($U^{-1} = U^\dagger$) linear operator $U$ on the corresponding Hilbert space. The Pauli group consists of the following canonical single-qubit gates (written wrt. basis $\{|0\rangle, |1\rangle\}$).

$$I = \begin{pmatrix} 1 & 0 \\ 0 & 1 \end{pmatrix}, \quad X = \begin{pmatrix} 0 & 1 \\ 1 & 0 \end{pmatrix}, \quad Y = \begin{pmatrix} 0 & -i \\ i & 0 \end{pmatrix} \quad \text{and} \quad Z = \begin{pmatrix} 1 & 0 \\ 0 & -1 \end{pmatrix}.$$

From their definition can be inferred that $X$ and $Z$ act as $X |b\rangle = |1 - b\rangle$ and $Z |b\rangle = (-1)^b |b\rangle$ on the qubit state. The single-qubit Pauli group generalizes to the $n$-qubit Pauli group $\mathcal{P}_n$, which consists of tensor products of single-qubit Pauli gates. Other useful quantum gates are the single-qubit Hadamard gate $H$ and phase gate $S$, which act as $H |b\rangle = (|0\rangle + (-1)^b |1\rangle)/\sqrt{2}$ and $S |b\rangle = i^b |b\rangle$ respectively. The canonical two-qubit gate is the controlled-NOT (CNOT), which operates on one target qubit conditioned on one control qubit by $|x, y\rangle \mapsto |x, x \oplus y\rangle$. For our purposes, a quantum circuit is a visual representation of sequence of quantum gates $[U_1, U_2, \cdots, U_k]$ applied left-to-right, and is thus associated with the quantum operation $U = U_k U_{k-1} \cdots U_1$.

## A.2 STABILIZER STATES

We say that an element $\pi \in \mathcal{P}_n$ stabilizes state $|\psi\rangle$ if $\pi |\psi\rangle = |\psi\rangle$. The set of stabilizers of a state comprises its stabilizer group (generated by at most $n$ elements). A state is a *stabilizer state* iff its stabilizer group is generated by $n$ elements. Conversely, every $\mathbb{S}$ uniquely determines a corresponding stabilizer state $|\psi\rangle$ as the simultaneous eigenstate with eigenvalue 1, $g |\psi\rangle = |\psi\rangle$, $\forall g \in \mathbb{S}$. Since a stabilizer group generator $\in \mathcal{P}_n$ can be represented using $2n + 1$ bits, a stabilizer state $|\psi\rangle$ can be written using $n(2n + 1)$ bits.

Stabilizer states can also be characterized as the states that can be reached from the all-zeros state $|\mathbf{0}\rangle$ using *Clifford* circuits, i.e. unitaries that are a combination of $H$, $S$ and CNOT gates. Notably, the Pauli gates are Clifford unitaries. The action of each of these gates on a stabilizer state's bit-representation is simple, resulting in the efficient classical simulation of quantum computation exclusively with Clifford unitaries (Aaronson & Gottesman, 2004).

Despite this, preparing stabilizer states optimally remains a challenge, since the number of Clifford states grows rapidly as $2^{\mathcal{O}(n^2)}$. Known optimal implementations have been limited to 6 qubits (Bravyi et al., 2022). Further, the (anti-)commutation and involutory properties of Clifford gates make it harder to reason about locally greedy search steps. We outline existing work towards stabilizer state preparation in Sec. 3.

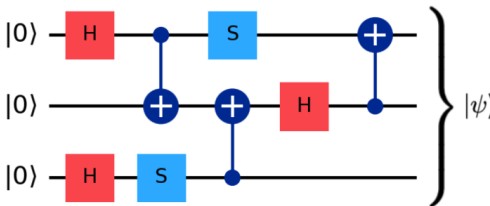

Figure 5: A Clifford circuit preparing stabilizer state $|\psi\rangle$. The $\oplus$-end of a CNOT gate denotes its target qubit.

Due to their simple mathematical structure and their ability to capture volume-law entanglement (where the entanglement content grows with the volume of the qubit lattice, i.e., $S \sim cn$ for a linear $n-$qubit chain) (Li et al., 2019), stabilizer states enjoy vast applicability. They have found immense use in the exploration of quantum information (Webb, 2016; Huang et al., 2020) and are also crucial for quantum error correction (QEC) (Gottesman, 1997; Nielsen & Chuang, 2010; Campbell et al., 2017; Ryan-Anderson et al., 2021). They are also applied beyond to measurement-based quantum computing (Raussendorf & Briegel, 2001; Patil & Guha, 2023), quantum-classical hybrid algorithms (Cheng et al., 2022; Ravi et al., 2022) and even ground-state physics (Sun et al., 2024).

### A.3 REINFORCEMENT LEARNING

In a Reinforcement Learning (RL) problem an agent learns through interactions with an environment to maximize its reward (Sutton & Barto, 2018). The environment is modeled as a Markov decision process, consisting of (a) a set $\mathcal{S}$ of states of the environment, (b) a set of actions $\mathcal{A}$ of the agent, (c) a transition function $p : \mathcal{S} \times \mathcal{A} \times \mathcal{S} \to [0, 1]$ where $p(s'|s, a)$ is the probability that the state of the environment will be $s'$ if the environment is in state $s$ and the agent takes action $a$, (d) a reward function $r : \mathcal{S} \times \mathcal{A} \times \mathcal{S} \to \mathbb{R}$ with $r(s, a, s')$ representing the *reward* that the agent receives from the environment for taking action $a$ from state $s$ and reaching state $s'$, and (e) a set $\mathcal{T} \subset \mathcal{S}$ of terminal states. The interaction between agent and environment stops on reaching a terminal state or exceeding a maximum number $T$ of actions without reaching a terminal state.

A policy of an RL agent is a function, $\pi : \mathcal{S} \times \mathcal{A} \to [0, 1]$ with $\pi(a|s)$ being the probability that the agent will take action $a$ when in state $s$. A trace $\tau$ of $\pi$ is a tuple of alternating states and actions, starting and ending in a state: $\tau = (s_0, a_0, s_1, \cdots, a_{T-1}, s_T)$. A policy $\pi$ along with a distribution $\mu$ over possible start states $s_0$ induces a distribution over traces, with $a_i \sim \pi(\cdot|s_i)$, $s_{i+1} \sim p(\cdot|s_i, a_i)$ for each $i$. The return of trace $\tau$ is defined by $G(\tau) := \sum_{i=0}^{T-1} \gamma^i r(s_i, a_i, s_{i+1})$, where $\gamma \in (0, 1)$ is the discount factor, describing the value of future actions in the present. The goal in RL is to find a policy $\pi^*$ that maximizes the expected return

$$J_\pi := \mathbb{E}_{\tau \sim (\mu, \pi)} [G(\tau)]. \tag{2}$$

Two key objects of interest in the search for such a policy are the value function $V^\pi(s) := \mathbb{E}_{\tau \sim \pi | s_0 = s} [G(\tau)]$ and the $Q$-function $Q^\pi(s, a) := \mathbb{E}_{\tau \sim \pi | s_0 = s, a_0 = a} [G(\tau)]$. An associated function is the advantage function, denoting how much better a particular action is w.r.t the average:

$$A^\pi(s, a) := Q^\pi(s, a) - V^\pi(s).$$

We use Proximal Policy Optimization (PPO) throughout our experiments. PPO (Schulman et al., 2017) is a reinforcement learning algorithm from the class of actir-critic algorithms designed to improve stability and efficiency in policy optimization. A running policy function $\pi$ (parameterized by $\theta$) and value function $V$ (parameterized by $\phi$) are maintained, typically as neural networks. Multiple agents gather experience by taking actions in the environment, according to current policy $\pi$. Concurrently, advantages $A^\pi(s, a)$ are estimated, approximating $Q^\pi(s, a)$ and $V^\pi(s)$ using sample averages over the experiences collected. In practice, one replaces advantages by generalized advantages, exponentially-weighted linear combinations of the advantages along a trace, which yield more robust estimates (Schulman et al., 2018).

Once sufficiently many steps are collected, we perform several optimization steps. Each optimization step starts by sampling a minibatch $\mathcal{D} = \{(s_i, a_i, \widehat{A}_i)\}_i$, where $\widehat{A}_i$ is the advantage estimate, from the experience pool $\mathcal{E}$. We next compute the policy objective of PPO, which can be viewed as a simplified alternative to the objective in Trust Region Policy Optimization (Schulman et al., 2015):

$$\mathcal{J}^{\text{CLIP}}(\theta) = \mathbb{E}_{(s_i, a_i, \widehat{A}_i) \sim \mathcal{E}} \left[ \min \left( r(\theta) \widehat{A}_i, \text{clip}\left( r(\theta), 1 - \epsilon, 1 + \epsilon \right) \widehat{A}_i \right) \right] \tag{3}$$

where $r_i(\theta) = \pi_\theta(a_i|s_i)/\pi_{\theta_{\text{old}}}(a_i|s_i)$ denotes the probability ratio between new and old policies with respective parameters $\theta$ and $\theta_{\text{old}}$. The clip function is defined for $a < b$ by $\text{clip}(x, a, b) = \max(a, \min(x, b))$. $\epsilon$ is the clipping hyperparameter; clipping ensures that the new policy does not deviate significantly from the old policy, thereby providing more stable learning. The gradient of the objective is computed and the parameters $\theta$ updated by gradient ascent. This completes one policy optimization step, and the process is now repeated, starting with sampling a new minibatch. Along with the policy objective, the value function is trained via the (clipped) value loss defined by

$$\mathcal{L}^{\text{value}}(\phi) = \mathbb{E}_{(s_i, a_i, \widehat{G}_i) \sim \mathcal{E}} \left[ \max \left( \left( \widehat{G}_i - V_\phi(s_i) \right)^2, \left( \widehat{G}_i - V_\phi^{\text{clip}}(s_i) \right)^2 \right) \right]. \tag{4}$$

Here, $V_\phi^{\text{clip}}(s_i) = V_{\phi_{\text{old}}}(s_i) + \text{clip}\left( V_\phi(s_i) - V_{\phi_{\text{old}}}(s_i), -\epsilon, \epsilon \right)$ stabilizes the update $\phi_{\text{old}} \to \phi$. $\widehat{G}_i$ refers to the cumulative reward obtained starting from $(s_i, a_i)$, estimated from the trace containing the step $(s_i, a_i)$. Finally, to encourage exploration, an entropy term is also included as part of the policy objective,

$$\mathcal{J}^{\mathcal{H}}(\theta) = -\mathbb{E}_{s_i \sim \mathcal{E}} \left[ \mathcal{H}(\pi_\theta(\cdot \mid s_i)) \right], \tag{5}$$

here $\mathcal{H}(\pi_\theta(\cdot \mid s_i))$ represents the *entropy* of the policy in state $s_i$.

# B    PROOFS

## B.1    MGR RETURN

We formally prove the following proposition.

**Proposition 4.** *Let $\Phi$ be an arbitrary potential function on state space $\mathcal{S}$, and $\gamma$ be the discount parameter. Consider the following MGR reward function $r$: Given $k \geq 0$ and $k$-step trajectory $\tau = (s_0, a_0, s_1, \cdots, s_k, a_k, s_{k+1})$. Letting $M_k = \max_{0 \leq i \leq k} \Phi(s_i)$, we set*

$$r(s_k, a_k, s_{k+1}) := \begin{cases} \gamma \Phi(s_{k+1}) - M_k & \Phi(s_{k+1}) > M_k \\ (\gamma - 1) M_k & \text{otherwise} \end{cases}.$$

*Consider a trajectory $\tau = (s_0, a_0, \cdots, s_{T-1}, a_{T-1}, s_T)$ that ran for $T$ steps. Then we have*

$$G(\tau) = \sum_{i=0}^{T-1} \gamma^i r(s_i, a_i, s_{i+1}) = \gamma^T \Phi^* - \Phi(s_0),$$

*where $\Phi^* = \max\{\Phi(s_i) \mid 0 \leq i \leq T\}$.*

*Proof.* Fix step $k$, and let $r_k := r(s_k, a_k, s_{k+1})$. If $\Phi(s_{k+1}) > M_k$, we have $M_{k+1} = \Phi(s_{k+1})$ and the reward $r_k = \gamma \Phi(s_{k+1}) - M_k = \gamma M_{k+1} - M_k$. Otherwise, $M_{k+1} = M_k$ and in this case, $r_k = (\gamma - 1) M_k = \gamma M_{k+1} - M_k$. It follows that

$$G(\tau) = \sum_{i=0}^{T-1} \gamma^i r_i = \sum_{i=0}^{T-1} \gamma^{i+1} M_{i+1} - \sum_{i=0}^{T-1} \gamma^i M_i = \gamma^T M_T - M_0.$$

We finish by noting that $M_T = \Phi^*$ and $M_0 = \Phi(s_0)$. $\qquad\square$

## C    GENERALIZATION BOUNDS FOR SUCCESS PROBABILITY

We use concentration to establish lower bounds on the probability $p$ of successfully preparing a uniformly sampled $n$-qubit stabilizer state.

**Proposition 5.** *Let* $\mathcal{A}$ *be a state preparation agent, taking a state* $|\psi\rangle$ *as input and outputting a circuit* $\mathcal{A}_{|\psi\rangle}$. *Let* $U(\mathcal{S}_n)$ *be the uniform distribution over* $n$-qubit stabilizer states, and define $X$ to be the random variable over $U(\mathcal{S}_n)$ by

$$X(|\psi\rangle) = \begin{cases} 1 & \mathcal{F}\left(A_{|\psi\rangle} |\psi_0\rangle , |\psi\rangle\right) = 1, \\ 0 & \text{otherwise.} \end{cases}$$

Now suppose that $|\psi_1\rangle, |\psi_2\rangle, \cdots, |\psi_N\rangle$ are sampled i.i.d $\sim U(\mathcal{S}_n)$. Let $X_i = X(|\psi_i\rangle)$ and define empirical mean $\bar{X} = \frac{1}{N} \sum_i X_i$.

Let $\varepsilon, \delta > 0$. Then with probability at least $1 - \delta$ over the choice of samples $(X_1, \cdots, X_N)$,

$$\mathbb{E}[X] \geq \bar{X} - \varepsilon$$

whenever

$$N \geq \frac{1}{2\varepsilon^2} \log \frac{1}{\delta}.$$

*Proof.* Denote the distribution of $X$ by $P$, and let $p := \mathbb{E}[X]$. By sampling uniformly random states, we essentially pick a sample $X_1, \cdots, X_n$ independently and identically distributed according to $P$. Since $0 \leq X \leq 1$, it follows by Hoeffding's inequality that

$$\Pr\left(\bar{X} \geq \mathbb{E}[X] + \varepsilon\right) \leq e^{-2N\varepsilon^2}.$$

The right-hand side is at most $\delta > 0$ whenever $N \geq \frac{1}{2\varepsilon^2} \log \frac{1}{\delta}$, so for such $N$,

$$\Pr\left(\mathbb{E}[X] \geq \bar{X} - \varepsilon\right) = \Pr\left(\bar{X} \leq \mathbb{E}[X] + \varepsilon\right) \geq 1 - \delta$$

as required.

$\square$

## D    ADDITIONAL EXPERIMENTS AND HYPER-PARAMETERS

### D.1    TWO-QUBIT GATE COUNT

We detail here an experiment that is not directly related to the problem that we attack, to which our agents continue to provide a good answer despite never being biased to do so.

**CNOT gate count**. Our metric for circuit size is the total number of gates, with both one and two-qubit gates counted as one unit each. However, since two-qubit gates are often noisier than single-qubit gates, we check our agents to examine the CNOT count, to see if we receive an additional benefit of smaller CNOT counts for free.

To this end, we benchmark our trained agents using the CNOT gate count as the metric of performance. Note that our agents are never explicitly trained to minimize two-qubit gates, and are trained with one and two-qubit gates placed on an equal footing. However, as our experiments on entanglement dynamics show (see Fig. 3(b)), the agent's actions do not display much redundancy and monotonically increase entanglement; one could expect good usage of the entangling CNOT gate. The results are shown in Tab. D.1. Note that the two baseline methods use the SWAP gate in addition to CNOT.

Tab. D.1 shows that we perform well, sometimes better than the optimized Bravyi et al. (2021) algorithm with respect to CNOT gates, despite having given no bias towards minimizing the number of two-qubit gates. This further emphasizes our efficiency in zero-shot state preparation.

Table 3: Average number of CNOT gates (↓) used by different algorithms across 200 randomly sampled uniform stabilizer states.

| Two-qubit gate count →) | 5-qubit | 6-qubit | 7-qubit | 9-qubit |
|---|---|---|---|---|
| Aaronson & Gottesman (2004) | $9.12 \pm 3.29$ | $14.92 \pm 3.85$ | $21.34 \pm 4.16$ | $38.22 \pm 5.46$ |
| Bravyi et al. (2021) | $7.56 \pm 2.46$ | $11.89 \pm 2.81$ | $\mathbf{16.30} \pm 2.86$ | $\mathbf{26.46} \pm 3.17$ |
| RL (linear connectivity) | $10.16 \pm 4.16$ | $14.50 \pm 7.34$ | $18.44 \pm 4.10$ | - |
| RL (full connectivity) | $\mathbf{6.08} \pm 2.45$ | $\mathbf{9.13} \pm 2.28$ | $19.52 \pm 7.71$ | $34.20 \pm 13.48$ |

Table 4: The list of PPO Hyperparameters that were tuned for agent preparation.

| Hyperparameter | Value |
|---|---|
| Learning rate (policy) | 0.0003 |
| Learning rate (value) | 0.0005 |
| Num. optimization epochs | 8 |
| Minibatch size | 256 |
| Discount ($\gamma$) | 0.99, 0.9 (9-qubit) |
| GAE parameter ($\lambda$) | 0.95 |
| policy_optimization_epochs | 8 |
| policy_clip_range | 0.2 |
| value_optimization_epochs | 8 |
| value_clip_range | $\infty$ |
| entropy_loss_weight | 0.01 |

## D.2 HYPER-PARAMETERS

We describe additional hyper-parameters part of the PPO algorithm (Schulman et al., 2017) that we used to train our agents.

All policy and value networks used had two hidden layers of $512$ nodes each.

