# OpenReview forum: "Train once and generalize: Zero-shot quantum state preparation with RL"
_ICLR.cc/2025/Conference — Submitted to ICLR 2025_

### Official Review · Reviewer_crcX · 2024-11-03

**Soundness:** 2
**Presentation:** 2
**Contribution:** 1
**Rating:** 3
**Confidence:** 4

**Summary:**

This work utilizes a reinforcement learning PPO agent to solve the quantum state preparation problem. The key idea involves a potential based reward that is based on goalposts - earlier fidelity penalties are weighted less to encourage exploration and the preparation of partially suboptimal but intermediately necessary states. Training is conducted on stabilizer circuits, which can be completely classically simulated and only involve the H, CNOT and phase gate variants, the placement of which are used as actions. To reduce the set of possible end-target states the RL agent trains to reverse the state preparation, starting from the target state towards the |0> state. After training a one-shot model is evaluated on a study of random target states and a set of states relevant for quantum error correction is shown to yield circuits preparing valid states with fewer gates than analytical models. An analysis on the entangling entropy is given empirically and a upper bound is approximated via Monte-Carlo estimations.

**Strengths:**

The ideas of this paper are presented understandable and mostly concise, the format is clean and the structure is logical. The background is explained very extensively. While I think the application of RL for stabilizer states is has fundamental issues, see below, I find the concept of reverse construction for a fixed end-state an interesting idea. I also think the extra effort for the study of entangling entropy and the attempt at a generalization bound, albeit quite loosely estimated, deserve praise and should be more common in empirical QRL papers.

**Weaknesses:**

Nonetheless, I think this paper has three major issues:

- While fundamental explanations are generally good, the first 5 1/2 pages are simply explanation of fundamental QC / RL basics (excluding Figure 2). The first original section starts at the bottom of page 6, which is not efficient use of space. Much of Chapter 2 could be formulated much more concise or moved to an appendix. In general the information and result density is rather light for a A* publication with 10 pages. On the other hand the actual RL algorithm realization, and the training details should be made much more clear. From what I understand the observation is the whole stabilizer tableau (flattened probably?) and the actions are discrete choices over the set of all applicable gate-actions - or gate-target pairs for two-qubit gates. I am not clear if the addition of g_i in L343 is included somehow, or how often the actions are chosen. Once for each qubit for every layer? Some of the training details like hyperparameters are in the appendix, but are critical information for a self-contained paper and should be moved to the main-paper. (E.g. the training episodes, the max-T curriculum for increasing n, the fidelity relaxation and the fact that the otherwise-case of the reward was not even used as described in the paper.)
- The goalpost potential-reward as part of the contribution feels quite over-engineered and I’d argue this work is lacking a critical evaluation in the RL context. Similarly, while the reverse generation thought is interesting, without a comparison to the forward-generation training (zero to target) I find it hard to judge any significance. A study of the generation with a simpler fidelity based reward as is found in the literature (even the ones cited here, Kölle et al., Gabor et al., Wu et al.) would have helped, as would a test with a simpler, fixed step-based regularizer instead of the goalpost. A direct comparison of ‘backward’ vs. ‘forward’ generation training would also help support the claims made here.
- Finally, I think the RL application for stabilizer circuits in particular has fundamental issues. While they are indeed used in QEC, the point and the reason for that is, that they are completely and very efficiently classically simulated. (The abstract of Aaronson & Gottesman mentions thousands of qubits.) In other words, any efficiency gain w.r.t. circuit size, generalization ability or else, could also be classically brute-forced. Without application to circuits with unsimulatable quantum-effects (including rotation gates, e.g.) efficient one-shot generalization is an interesting insight, but lacking overall impact in terms of QC relevancy.
---
Minor Issues:
- L256 Typo: parameters: [a] the target …
- Inconsistent use of ‘Fig.’ and ‘Figure’ references.

**Questions:**

- This max { } function notation is confusing to me, L323 (e.g.) uses max{a : b}, L324 uses max{a | b}? How is this supposed to be read and understood?
- Why was n=9 picked as the benchmark and not something higher, since stabilizer circuits can be so efficiently computed? L517 mentions that the he 9-qubit agent was not trained to convergence, why is that?
- How were the observation and action choices implemented in detail?

---

> ### Author Response · Authors · 2024-11-27
> **Rebuttal by authors [1/2]**
>
> We sincerely appreciate your insightful and valuable feedback. It has been carefully incorporated into the revised manuscript.
>
> > While fundamental explanations are generally good, the first 5 1/2 pages are ... training details should be made much more clear.
>
> Thank you for the valuable suggestions, it is much appreciated. According to your feedback, we have re-organized the paper to explain the methods, reward function, full experimental setup and descriptions of the experiments in much more clarity, and have moved the fundamental parts to the appendix. All experiment scenarios and assumptions have been motivated and remarked on at the start of the section. We make clear the various benchmarks, tests and results we obtain with the seven prepared agents. We also improve our contrast with previous work and include some new works in the "related works" section. We also present more motivation for our reward and a numerical comparison with other reward functions. The methods now span pages 4-6, and the experiments pages 7-10. We hope that the issues on presentation and results have been satisfactorily addressed in the revised document.
>
> > From what I understand the observation ... for two-qubit gates
>
> Yes, these are indeed true. Our apologies for not explicitly making this clear, we have done so in the revised paper.
>
> > I am not clear if the addition of $g_i$ ... for every layer?
>
> The quantity $g_i$ is only used to simplify the proof of the expression for cumulative reward and is not required in the algorithm or otherwise.
> The agent chooses one action (either picks a qubit and a single-qubit gate to apply, or a pair of control-target qubits and a two-qubit gate to apply) at each step. In this work, we aimed to optimize circuit size in terms of number of gates rather than circuit depth. We have clarified this in the revision, please see Table 1.
>
> >  Some of the training details like ... as described in the paper.
>
> Thank you for the valuable suggestion. We concur in that these are important details crucial to the success of the experiments, and have added these to the main paper. The training episodes and max-T value for episode truncation have been moved into the main paper. The fact that the otherwise-case of the reward was not used is now part of Sec. 4.2 on the reward, where we also provide training graphs justifying our choice -- it learns faster, partially because of the increased freedom to choose gates that do not increase fidelity, arising from a smaller punishment (exactly $-\alpha$).
>
> > The goalpost potential-reward ... fixed step-based regularizer instead of the goalpost.
>
> Thank you for this crucial suggestion. Accordingly, we have made a study of the different rewards typically used and also bench-marked performance. We bench-mark our reward against simpler rewards used earlier in the literature, and show that our reward is singularly able to guide the agent towards a good policy. Please see Sec. 4.2 and Fig. 2 of the revised manuscript.
>
> > Similarly, while the reverse ... would also help support the claims made here.
>
> We would like to point out that the most important difference between the two would be the generalization -- forward-generation is limited to preparing one state at a time, since the target state in reinforcement learning is constant. Backward generation, on the other hand, learns to prepare arbitrary states during a single training regimen, including states unseen during the course of training. We exploit that the starting state in RL need not be constant and could vary throughout the training. In fact, we show that our agents generalize to at least $95$\% of the state space. This implies that even for $7$ qubits, backward preparation on $130$K episodes, each of at most 80 steps -- 10.4M training steps in total -- is able to prepare at least 95\% of the 81.3 billion $7$-qubit states, i.e. at least 77.2 billion states. This is a massive generalization over forward training, where training for such a number of states is infeasible. With $9$ qubits, the generalization soars to $4 \times 10^{16}$ states. Thus, the benefit of extensive generalization offers a much more scalable and comprehensive tool for state preparation.
>
> > Finally, I think the RL ... QC relevancy.
>
> We would like to point out that the preparation of stabilizer states is still a hard task despite classical simulability. Infact, the classical simulability is a numerical convenience used to train these algorithms as we clarify in detail in the general response. Moreover, we stress that the reason for their use in QEC is _not_ classical simulability, it is the well-defined nature of codes one can define using Clifford circuits and do error correction.
>
> > This max ... read and understood?
>
> We apologize for this oversight: both of them were supposed to represent the maximum of set $ \\{ a:b\\} \equiv \\{a|b\\}$, written inconsistently in set-builder form. We have made the exposition much more clearer in the revised paper.

---

> ### Author Response · Authors · 2024-11-27
> **Rebuttal by authors [2/2]**
>
> > Why was n=9 ... computed?
>
> We agree that stabilizer circuits can be efficiently _simulated_, however, as mentioned above, this does not imply that it is easy to prepare efficient circuits that prepare stabilizer states. Preparing an efficient circuit corresponds to finding a short circuit from among an exponentially large space of possibilities, i.e. set of Clifford circuits preparing the desired state. This is a difficult optimization problem with no optimal scalable solution, even allowing for re-training to prepare each particular state. Please see the general response for further details about the same.
>
> > L517 mentions ... why is that?
>
> This remark was a mistake on our part; the agent was trained to successful convergence, we simply did not recognize it. We apologize for the confusion. Given that the average return at the time was (slightly) negative, we believed the solution was currently sub-optimal and that the reward would eventually rise, leading to the comment in the paper. However, later analysis showed that the negative reward is expected, since we had chosen $\gamma = 0.9$ for this experiment and the average episode length was $T > 40$, the constant term $-\alpha$ outweighed benefits. In particular, the sum of the positive $\gamma^T\Phi^*$ and $\gamma^j\max\{\Phi(s_i)\,:\,1\leq i\leq j\}$ terms were outweighed by the $-\alpha$ term, i.e. $-\sum_{i=0}^{T-1}\alpha\gamma^i$, leading to net (slightly) negative reward. This was not the case with the other experiments where $\gamma = 0.99$, which yielded positive reward upon successful convergence. So we concluded, forgetting that $\gamma$ was 0.9 for the 9-qubit case, that the agent was not done and would later perform better, though an increase in training time did not yield a substantial improvement -- which is obvious.
>
> TL;DR we simply did not recognize that our agent had actually successfully converged to a good policy. In fact, we show in the revised manuscript that the $9$-qubit agent generalizes to at least $95$\% of the $9$-qubit stabilizer state space, a massive generalization.
>
> > How were the observation ... in detail?
>
> We apologize for not making this clear in the original draft. The observation space is the set of $n$-qubit stabilizer states, with each state being represented as its stabilizer tableau [A]. The action space is a collection of $n$-qubit unitaries induced from the allowed gate-set and the qubit connectivity. For example, say $n = 3$ and the connectivity graph has edges $\\{q_1, q_2\\}, \\{q_2, q_3\\}$ and $\\{q_3, q_1\\}$. In the revised manuscript, we have addressed this in detail along with the other concerns.
>
> [A] Scott Aaronson and Daniel Gottesman. Improved simulation of stabilizer circuits. Physical Review A, 70(5), November 2004.
>
> ---
>
> We deeply value the time and effort you have invested in providing detailed feedback, as well as the excellent suggestions that have strengthened this work. We hope our responses and the new evidence we have shared have resolved your concerns, and would greatly appreciate if this could be reflected in your evaluation. Please do not hesitate to reach out with any further comments or questions.

---

> ### Author Response · Authors · 2024-12-01
> **Discussion phase ending soon - awaiting your response**
>
> Dear Reviewer,
>
> Thank you for your detailed feedback. We have worked hard to address your concerns by re-organizing the paper to explain the methods and experiments with more motivation and clarity, also providing the full experimental setup and motivation before each experiment. We have also added theoretical and numerical evidence to support our reward function, along with a training figure supporting its improved and sample-efficient performance against typically used reward functions. We have also addressed the issues pointed out about stabilizer state preparation, and clarified convergence of the 9-qubit agent.
>
> Since the author-reviewer discussion period ends soon, we shall be grateful if you could consider upgrading your score to reflect the overall improvement due to these enhancements. We also welcome any further questions, concerns, or suggestions. Many thanks!
>
> Best regards,
>
> The authors

---

### Official Review · Reviewer_GvX9 · 2024-11-04

**Soundness:** 2
**Presentation:** 2
**Contribution:** 2
**Rating:** 3
**Confidence:** 4

**Summary:**

The manuscript presents a deep reinforcement learning approach for quantum state preparation (QSP) . The authors design a novel reward function with guarantees that significantly scales beyond previous works, allowing for generalization to unseen states. They demonstrate their method on stabilizer states up to nine qubits, achieving state-of-the-art performance by preparing target states with varying degrees of entanglement.

**Strengths:**

- This manuscript introduces a novel novel reward function with provable guarantees, which adds a level of theoretical robustness to the approach.
- The proposed method's success on stabilizer states, which are crucial for quantum error correction and other quantum information processes, suggests its applicability.
- The paper demonstrates that their method generates circuits that are up to 60% shorter than those produced by existing algorithms in some cases, which is an improvement in efficiency.

**Weaknesses:**

- I believe the term "zero-shot" in the title of this paper is misleading.Zero-shot learning (ZSL) is a problem setup in deep learning where, at test time, a learner observes samples from classes which were **not** observed during training, and needs to predict the class that they belong to. However, in the context of this paper, training is conducted on a set $\mathcal S$ of $n$-qubit states. From my perspective, this setup aligns more with traditional supervised learning rather than what is commonly referred to as "zero-shot" learning.
- The authors claim that "The key idea that allows us to obtain zero-shot inference is the following: a sequence of actions starting in state |ψ⟩ leading to the state |0⟩ yields the inverse of a circuit preparing |ψ⟩ starting from |0⟩". However, I believe this concept is not original to this paper. For example, a similar idea was proposed in [1]. Yet, the authors have failed to cite it.
- I doubt the authors overstated their results. It's important to distinguish between the preparation of arbitrary quantum states and the preparation of arbitrary stabilizer states. The latter is a more specialized and arguably less challenging problem compared to the general case of arbitrary state preparation. Stabilizer states, due to their specific properties and structure, may be more amenable to efficient preparation methods, which might not directly translate to the broader and more complex task of preparing any arbitrary quantum state.

[1] Huang, Hsin-Yuan, et al. "Learning shallow quantum circuits." *Proceedings of the 56th Annual ACM Symposium on Theory of Computing*. 2024.

**Questions:**

Please see the "Weaknesses" above.

---

> ### Author Response · Authors · 2024-11-27
> **Rebuttal by authors**
>
> Thank you for your valuable inputs and feedback. We believe the answers below will help clarify your concerns.
> > I believe the term "zero-shot" ... what is commonly referred to as "zero-shot" learning.
>
> Thank you for this fair concern. Indeed, as is correctly pointed out, ZSL is typically used to indicate the classification
> of a sample whose class has not been observed during training. We were inspired by this idea, put in the context of state preparation: the preparation of a state whose circuit has not been observed during training was to count as zero-shot. Since this is not standard, we have clarified our definition of zero-shot in the introduction of the revised manuscript. To quote, “An agent that does not need re-training to prepare unseen states will be called zero-shot in this work, to emphasize successful generalization to states not seen during training."
>
> > The authors claim that ...  have failed to cite it.
>
> Our apologies for this oversight. We would like to clarify that while [1] indeed uses a similar trick of inversions, their protocol is specific to local inversions and is only applicable to geometrically-local shallow circuits. Moreover, the algorithm in [1] proceeds by learning such inversions given each new input state. Whereas, we perform the inversion trick only during the training phase, and can subsequently generalise to almost all other stabilizer states. Nonetheless, the approach of [1] is indeed important, and is very relevant to the RL scenario where one would train a local RL agent to enable further scalability. Thus, we have included work using this approach in the revised manuscript, and have outlined this important direction of future work combining the techniques in [1] with our RL framework.
>
> > I doubt the authors ... complex task of preparing any arbitrary quantum state.
>
> We greatly appreciate your feedback, and have acted on this important distinction. Specifically, we now clearly mention that we prepare stabilizer states in the abstract and introduction along with the already existing clarification in the main text. While the intuition that stabilizer state preparation is arguably less complex than the fully general problem is indeed true, this work addresses in full generality the problem of stabilizer state preparation and the relevance and promise of reinforcement learning methods for the same. We further refer to the general response for extended clarification on this issue.
>
> ---
>
> We deeply value the time and effort you have invested in providing detailed feedback, as well as the excellent suggestions that have strengthened this work. We hope our responses and the new evidence we have shared have resolved your concerns, and would greatly appreciate if this could be reflected in your evaluation. Please do not hesitate to reach out with any further comments or questions.

---

> ### Author Response · Authors · 2024-12-01
> **Discussion phase ending soon - awaiting your response**
>
> Dear Reviewer,
>
> Thank you for your valuable feedback. We have made our best attempt to address your concerns in our response, by clarifying our definition of zero-shot, incorporating and contrasting relevant work that also use an inverse-preparation-style idea in the revised manuscript, and addressing our claim of preparing arbitrary states. Also, we have improved our presentation and greatly strengthened our theoretical guarantees with concentration. We also clarify the notion of zero-shot used, 9-qubit agent convergence, and elaborate on the definitions requested.
>
> Since the author-reviewer discussion period ends soon, we shall be grateful if you could consider upgrading your score to reflect the overall improvement due to these enhancements. We also welcome any further questions, concerns, or suggestions. Many thanks!
>
> Best regards,
>
> The authors

---

### Official Review · Reviewer_hnhQ · 2024-11-04

**Soundness:** 3
**Presentation:** 3
**Contribution:** 3
**Rating:** 6
**Confidence:** 4

**Summary:**

This manuscript studied the problem of quantum state preparation (QSP). While theoretical lower bounds for the required operations have been established, efficiently approximating quantum states on near-term quantum computers remains an open question. The authors proposed a reinforcement learning (RL) algorithm for QSP, focusing on the preparation of stabilizer states— a set of states of significant practical interest—using Clifford gates. The authors introduced a unified target state by replacing $|\psi\rangle$ with $|0\rangle$ and approximating backwards, referred as "Zero-shot" in the manuscript. Additionally, the authors proposed a moving-goalpost reward (MGR) function that aligns the maximum cumulative reward with the highest final fidelity. On the experimental side, the authors conducted experiments with up to 9 qubits, with and without connectivity restrictions, achieving better performance in terms of gate count compared to referenced algorithms. Theoretically, they proved a loose lower bound on the probability of generalization success.

**Strengths:**

1. The problem studied is highly relevant to the quantum computing community. The manuscript presented a RL algorithm to efficiently prepare stabilizer state, which is valuable for future research.

2.  The numerical results is promising compared to conventional algiorthm, even with connectivity constraints, showing the potential utility on near-term quantum computers.

**Weaknesses:**

1. The nanuscript lack discussing about relevant literatures. The "Zero-shot" trick is not new; amany existing works have adopted it in reinforcement learning for quantum circuit synthesis and compiling, with QSP being only a subset [1,2].

[1] Zhang, Yuan-Hang, et al. "Topological quantum compiling with reinforcement learning." Physical Review Letters 125.17 (2020): 170501.
[2] Qiuhao, Chen, et al. "Efficient and practical quantum compiler towards multi-qubit systems with deep reinforcement learning." Quantum Science and Technology (2024).

2. The claim in the abstract that "To our knowledge, this is the first work to prepare arbitrary stabilizer states on more than two qubits without re-training" is questionable. A recent work [3] also proposed a reinforcement learning algorithm for Clifford circuit synthesis.

[3] Kremer, David, et al. "Practical and efficient quantum circuit synthesis and transpiling with Reinforcement Learning." arXiv preprint arXiv:2405.13196 (2024).

3. The authors established a loose lower bound for the probability of generalization success (as indicated in the last equation of the main text). In Table 2, the probability decreases with qubit size, raising concerns about the generalizability of the proposed algorithm. Additionally, in Line 517, the authors stated, "We did not train the 9-qubit agent to convergence...," which raises further concerns about the trainability of the proposed RL algorithm.

**Questions:**

1. The term "Zero-shot" is ambiguous. Typically, "zero-shot" refers to training without class labels. In the manuscript, the authors seem to be referring to the unified target trick, which should be clarified.

2. In the introduction, the authors claim that their RL method is applicable to "arbitrary states." However, the states studied are specifically stabilizer states. The authors should either correct this claim or provide results for additional categories of states.

3. How are the experimental results related to the theoretical lower bound of generalization? The authors should consider displaying the success probability achieved during their experiments.

4, What are the definitions of $q$ and $N$ in Lines 517-518? Additionally, the last equation should be numbered for clarity.

---

> ### Author Response · Authors · 2024-11-27
> **Rebuttal by Authors (1/2)**
>
> Thank you so much for the very insightful and important feedback. We have incorporated your feedback into the revised manuscript.
>
> > The manuscript lack discussing about relevant literatures. The "Zero-shot" trick is not new; many existing works have adopted it in reinforcement learning for quantum circuit synthesis and compiling, with QSP being only a subset [1,2].
>
> Thank you for bringing this to our attention. We have included these and other references that use a "zero-shot" trick for state preparation, in our revised paper.
>
> > The claim in the abstract that "To our knowledge, this is the first work to prepare arbitrary stabilizer states on more than two qubits without re-training" is questionable. A recent work [3] also proposed a reinforcement learning algorithm for Clifford circuit synthesis.
>
> Many apologies for missing this work. Indeed, it also proposes a framework for Clifford synthesis, and does not require re-training. We have included it as part of related work, and will remove the comment from the abstract. We also contrast our work with that of [3] in the revised manuscript to make our contributions clear.
>
> Although our framework also uses a similar idea, the key driver is actually our novel reward function, which yields our scalable and sample-efficient agents. The MGR reward enables us to scale up beyond previous works, while staying zero-shot. While Kremer et al [3] are also able to scale up stabilizer preparation to 11 qubits, we are 50X-100X more sample-efficient, requiring only 10-20M samples rather than 1B+ samples as used in [3], to train our agents. Further, we are also able to prove that our agents generalize to $95$\% of the state space, which is a massive generalization many orders larger than the training set, that we believe not been shown previously. Thank you for this valuable reference.
>
> > The authors established a loose lower bound for the probability of generalization success (as indicated in the last equation of the main text). In Table 2, the probability decreases with qubit size, raising concerns about the generalizability of the proposed algorithm.
>
> Thank you for your careful reading, and for pointing this out. Based on your feedback, we have now been able to obtain a much tighter bound for the generalization success, which asserts that with probability at least $1-10^{-10} \approx 1$, our agents generalize to at least $95$\% of the state space, for each $n = 5, 6, 7, 9$ without exception. We prove this using concentration bounds and a proof is detailed in Appendix C.
>
> > Additionally, in Line 517, the authors stated, "We did not train the 9-qubit agent to convergence...," which raises further concerns about the trainability of the proposed RL algorithm.
>
> We apologize for the confusion. We had actually trained it to convergence. Given that the average return at the time was (slightly) negative, we believed the solution was currently sub-optimal and that the reward would eventually rise, leading to the comment in the paper. However, later analysis showed that the negative reward is expected, since we had chosen $\gamma = 0.9$ for this experiment and the average episode length was $T > 40$, the constant term $-\alpha$ outweighed benefits. In particular, the sum of the positive $\gamma^T\Phi^*$ and $\gamma^j\max\{\Phi(s_i)\,:\,1\leq i\leq j\}$ terms were outweighed by the $-\alpha$ term, i.e. $-\sum_{i=0}^{T-1}\alpha\gamma^i$, leading to net (slightly) negative reward. This was not the case with the other experiments where $\gamma = 0.99$, which yielded positive reward upon successful convergence. So we concluded, forgetting that $\gamma$ was 0.9 for the 9-qubit case, that the agent was not done and would later perform better, though an increase in training time did not yield a substantial improvement -- which is obvious.
>
> TL;DR we simply did not recognize that our agent had actually successfully converged to a good policy. In fact, as mentioned above, we show in the revised manuscript that the $9$-qubit agent generalizes to at least $95$\% of the $(4.3\times 10^{16})$-sized $9$-qubit stabilizer state space -- many orders of generalization.

---

> ### Author Response · Authors · 2024-11-27
> **Rebuttal by Authors (2/2)**
>
> > The term "Zero-shot" is ambiguous. Typically, "zero-shot" refers to training without class labels. In the manuscript, the authors seem to be referring to the unified target trick, which should be clarified.
>
> Many thanks for your perspicacious remark. We agree that usage of "zero-shot" without some additional context could be confusing. Indeed, multiple zero-shot settings have received attention in the community such as predicting images from a class without having been presented any examples of that class during training, or more recently, in the context of so called 'zero-shot prompting' in LLMs. We were inspired by this same idea, put in the context of state preparation: the preparation of a state whose circuit has not been observed during training was to count as zero-shot. Since this is not standard, we have clarified our definition of zero-shot in the introduction of the revised manuscript. To quote, "*An agent that does not need re-training to prepare unseen states will be called zero-shot in this work, to emphasize successful generalization to states not seen during training.*"
>
> > In the introduction, the authors claim that their RL method is applicable to "arbitrary states." However, the states studied are specifically stabilizer states. The authors should either correct this claim or provide results for additional categories of states.
>
> Thank you for the opportunity to clarify this. We are currently working on the preparation of more general states using our framework. We have already used the same framework and reward function to prepare agents that successfully solve the zero-shot general state preparation problem for up to three qubits with $\varepsilon = 0.95$. However, as you correctly mention, the current work does not indicate this, since its focus is on stabilizer states -- we have corrected the claim to read "arbitrary *stabilizer* states".
>
> > How are the experimental results related to the theoretical lower bound of generalization? The authors should consider displaying the success probability achieved during their experiments.
>
> Thank you for bringing up this valuable point. In our experiments, the agent always succeeds -- all states are successfully prepared to a fidelity of $1.0$, and so success probability is $1$.
>
> The theoretical bounds provide rigorous insight into these observed results: large generalization provides evidence as to the empirical successful performance of the agent.
>
> As previously remarked, we were able to show much stronger bounds on generalization in the revised paper -- our agents provably generalize to at least $95$\% of the state space. This provides very strong theoretical evidence as to why our agents are always successful in preparation, as was suggested by the experiments.
>
> Thank you for this point -- it is indeed important to mention this connection between theory and experiment and we have done so in the revised manuscript.
>
> > What are the definitions of $q$ and $N$ in Lines 517-518? Additionally, the last equation should be numbered for clarity.
>
> We apologize for not making the definitions clear in the initial draft. $N$ is the number of states that we are guaranteed generalization to, and is computed as $N = pN^*$. Here $N^*$ is the total number of $n$-qubit stabilizer states and $p$ is (a lower bound on) the probability of successful preparation of a uniformly random stabilizer state. The quantity $q$ is the `generalization per training state seen' and is computed as $N/N_0$. Here $N_0$ is the total number of states seen during training and is equal to the number of training steps. The value $q$ then provides an insight into the amount of generalization one could expect *per state seen during training*.
>
> As remarked earlier, we proved a much tighter bound on $p$ while working on the revision. Our agents generalize to at least 95\% of the state space for each $n = 5, 6, 7, 9$. So $N = 0.95N^*$ and $q = N/N_0$ take values as shown:
>
> | Qubits   | N*                  | N                    | N₀                  | q                    |
> |----------|---------------------|----------------------|---------------------|----------------------|
> | 5 | $2.40 \times 10^6$   | $2.30 \times 10^6$   | $1.40$M  | 1.64                 |
> | 6 | $3.15 \times 10^8$   | $3.00 \times 10^8$   | $6.40$M  | 46.88                |
> | 7 | $8.13 \times 10^{10}$| $7.72 \times 10^{10}$| $10.4$M  | 7423.08              |
> | 9 | $4.38 \times 10^{16}$| $4.07 \times 10^{16}$| $22.9$M  | $1.77 \times 10^9$   |
>
> We hope this clarifies the meaning and purpose of $q$ and $N$.
>
> -------
> We greatly appreciate the detailed feedback and thoughtful suggestions that have helped enhance the presentation and overall quality of this work. We trust that our responses and the additional evidence provided address the concerns raised, and we would be grateful if this could be reflected in a revised evaluation. Should you have any further questions or ideas, we are more than happy to discuss them.

---

> ### Author Response · Authors · 2024-12-01
> **Discussion phase ending soon - awaiting your response**
>
> Dear Reviewer,
>
> Thank you for your detailed feedback. We have made our best attempt to address your concerns in our response, by clarifying our claims, incorporating and contrasting relevant work in the revised manuscript, and greatly strengthening our theoretical bounds to hold for every agent. We also clarify the notion of zero-shot used, 9-qubit agent convergence, and elaborate on the definitions requested.
>
> Since the author-reviewer discussion period ends soon, we shall be grateful if you could consider upgrading your score to reflect the overall improvement due to these enhancements. We also welcome any further questions, concerns, or suggestions. Many thanks!
>
> Best regards,
>
> The authors

---

> ### Comment · Reviewer_hnhQ · 2024-12-01
>
> I appreciate the authors’ efforts to address the concerns raised in the initial review. The responses provided clarifications on the generalizability of the method on unseen stabilizer states, the naming of the "zero-shot" term, and the issues related to convergence on 9-qubit tasks. These clarifications have improved the presentation of the work, and I have raised my evaluation regarding the presentation and contribution. However, I am unable to give a higher overall rating than my initial one, due to remained concerns about the generalizability to non-stabilizer states. Specifically, my concerns are as follows:
>
> 1. The authors' general response mentions that the evaluation of fidelity is based on stabilizer tableau for stabilizer states, which has a complexity of $O(n^2)$.  While this is feasible for stabilizer states, it may be difficult for non-stabilizer states. Evaluating the fidelity between two arbitrary states could prove to be significantly more challenging, especially considering the complexity of querying the target state.
>
> 2. The reward associated with deeper circuits may become negative, as observed in the 9-qubit tasks. For approximating general non-stabilizer states, the required circuit depth could grow substantially with the number of qubits. Would this lead to poor performance or a suboptimal learned policy?
>
> Additionally, the authors claim that the MGR reward improves sample efficiency in RL. Could the authors provide a comparison between MGR and the basic reward on the test set? While Figure 2 indicates that MGR might lead to faster convergence, the results on the 9-qubit tasks suggest that the RL model may not require convergence to succeed on the test set. This is just a suggestion to enhance the work's presentation, as having data samples on the order of billions to millions is already quite impressive compared to other literature.

---

> > ### Author Response · Authors · 2024-12-02
> > **Response by Authors**
> >
> > Thank you so much for your response and the very important questions. We hope the answers below will help clarify these questions.
> >
> > > The authors' general response mentions that the evaluation of fidelity is based on stabilizer tableau for stabilizer states, which has a complexity of $O(n^2)$. While this is feasible for stabilizer states, it may be difficult for non-stabilizer states. Evaluating the fidelity between two arbitrary states could prove to be significantly more challenging, especially considering the complexity of querying the target state.
> >
> > Thank you for this valid concern. Classical shadows can be used to show that a small number of single-qubit Pauli measurements suffice to accurately predict the fidelity of a state to any other [1]. We can use this to still provide the reward function. To decide termination of the episode, i.e., certify that the current state is $\varepsilon$-close to the fiducial $\ket{0}^{\otimes n}$, we use the method outlined in [2] that requires only $\mathcal O(n^2)$ single-qubit Pauli measurements and with high probability outputs $1$ iff the state is $\varepsilon$-close. It follows that we can replace the fidelity computation throughout the framework with these fast approximate methods, and implementing this is an immediate future work we are pursuing.
> >
> > [1] Huang et al, Predicting many properties of a quantum system from very few measurements. Nat. Phys. 16, 1050–1057 (2020).
> >
> > [2] Huang et al, Certifying almost all quantum states with few single-qubit measurements. arXiv:2404.07281 (2024).
> >
> > > The reward associated with deeper circuits may become negative, as observed in the 9-qubit tasks. For approximating general non-stabilizer states, the required circuit depth could grow substantially with the number of qubits. Would this lead to poor performance or a suboptimal learned policy?
> >
> > Thank you for raising this important point. Indeed, the required circuit size grows substantially with the number of qubits. However, performance depends only on whether the reward is able to guide the agent towards a good policy, and not on the sign of the reward. So far, we have already been able to successfully train an agent for $3$-qubit general state preparation to $95$\% fidelity with our reward. We believe that the main factor to trainability for the general problem is for the reward function to correctly capture the fidelity landscape for general states, and studying this is an important immediate future work.
> >
> > > Additionally, the authors claim that the MGR reward improves sample efficiency in RL. Could the authors provide a comparison between MGR and the basic reward on the test set? While Figure 2 indicates that MGR might lead to faster convergence, the results on the 9-qubit tasks suggest that the RL model may not require convergence to succeed on the test set. This is just a suggestion to enhance the work's presentation, as having data samples on the order of billions to millions is already quite impressive compared to other literature.
> >
> > Thank you for the suggestion. Actually, the $9$-qubit model did converge, as mentioned in the original rebuttal above. It is true that the RL model does not need full convergence to succeed on the test set. However, it still needs the average training fidelity to cross a certain threshold, since in all our experiments, the fidelities on the test set roughly match the average fidelity at the end of training. We present here the results of running the models of Figure 2 after the training period.
> >
> > | Model           | Average Fidelity (Mean ± Sigma) |
> > |------------------|----------------------------------|
> > | **INCR-PENALTY** | 0.0347 ± 0.0217                 |
> > | **INCR-VANILLA** | 0.0366 ± 0.0173                 |
> > | **POTENTIAL**    | 0.5 ± 0                         |
> > | **MGR-VANILLA**  | 1.0 ± 0                         |
> > | **MGR-OURS**     | 1.0 ± 0                         |
> >
> >
> > In particular, the other rewards are unable to learn or succeed on the test set.
> >
> > ----
> >
> > Thank you once again for the insightful questions. Please do not hesitate to reach out with any further comments or questions.

---

> > > ### Comment · Reviewer_hnhQ · 2024-12-03
> > >
> > > Thanks for the responses.
> > >
> > > "So far, we have already been able to successfully train an agent for $3$-qubit general state preparation to $95$% fidelity with our reward." I could not find relevant results related to this in the revision. Could you clarify if this result is included, and if so, where it can be found in the paper?
> > >
> > > I have another question about the sample complexity in each RL trajectory. Specifically, for a circuit of length $L$, measurement information is required for each instantaneous state along the trajectory, resulting in a total of $L + 1$ states for one sample. I believe this factor should be accounted for in the total sample complexity. Could you please clarify how this is handled?

---

> ### Author Response · Authors · 2024-12-03
> **Response by Authors**
>
> Thank you for the important questions.
>
> > "So far, we have already been able to successfully train an agent for $3$-qubit general state preparation to $95$\% fidelity with our reward." I could not find relevant results related to this in the revision. Could you clarify if this result is included, and if so, where it can be found in the paper?
>
> While we focused on stabilizer state preparation in this work, we found during our preliminary experiments during the discussion period that our framework and reward function generalize to the preparation of few qubit non-stabilizer states as well (as stated for $3$-qubit general state preparation). We will be sure to include these results in the final version.
>
> For completeness, here is the full experimental setup for $3$-qubit general state preparation (to $99$\% fidelity, in fact). The reward function is the same as used in the stabilizer experiments, i.e. **MGR-OURS**.
>
> | **Parameter**                     | **Value**                                                  |
> |------------------------------------|-----------------------------------------------------------|
> | **Gateset**                        | $H$, $S$, $Z$, $T$ single-qubit gates; CNOT gates (full connectivity) |
> | **Episodes**                       | 30,000                                                    |
> | **Steps per episode ($T^{\*}$)**      | 350                                                       |
> | **Policy and Value network sizes** | 3 hidden layers of size 512 each                          |
> | **Discount ($\gamma$)**            | 0.95                                                      |
>
> Testing was done on 1500 Haar-random $3$-qubit states, with every state reaching fidelity of $99$\%+ on the agent's first attempt. The average gate count was $278$, and average fidelity $0.9916 \pm 0.011$. 175 states needed at most $200$ gates, and another 325 between 200 and 250 gates.
>
> > I have another question about the sample complexity in each RL trajectory. Specifically, for a circuit of length $L$, measurement information is required for each instantaneous state along the trajectory, resulting in a total of $L + 1$ states for one sample. I believe this factor should be accounted for in the total sample complexity. Could you please clarify how this is handled?
>
> Thank you for raising this point. Indeed, the sample complexity of computing measurement information, followed by that of the computation of a certification of closeness to $\ket{0}^{\otimes n}$ for each step, add up to give $(L+1)\times C$ cost per trajectory. Here, $C$ is the sum of the sample complexities of both sets of measurements. Thank you once again; this is a very relevant and interesting point for future work using local Pauli measurements to represent the state.
>
> --------
> Thank you once again for the great questions. We welcome any further questions, concerns, or suggestions.

---

### Official Review · Reviewer_3SCv · 2024-11-04

**Soundness:** 3
**Presentation:** 3
**Contribution:** 3
**Rating:** 6
**Confidence:** 3

**Summary:**

This study presents a zero-shot reinforcement learning (RL) algorithm that incorporates a novel reward function specifically designed for quantum state preparation (QSP). The authors concentrate on stabilizer state preparation tasks involving up to 9 qubits. Notably, the proposed algorithm demonstrates strong generalization capabilities on unseen data, utilizing a training dataset that is 0.001% smaller than the total state space. Furthermore, the quantum circuits generated by this algorithm achieve a circuit size reduction of over 60% compared to existing methods.

**Strengths:**

1. The authors introduce a novel reward function that is inspired by the intrinsic nature of the QSP problem, integrating it into the RL framework to enhance performance in terms of circuit size reduction compared to existing algorithms.
2. The proposed algorithm demonstrates zero-shot generalization, significantly decreasing computational complexity compared to traditional RL algorithms, which require retraining for each new target state.
3. The authors provide a theoretical analysis of the generalization capabilities of the proposed algorithm.

**Weaknesses:**

1. As someone who is not well-versed in RL algorithms, I found it somewhat challenging to follow the overall algorithms framework. The authors present the general RL algorithms separately before introducing their specific formulation for QSP tasks along with the novel reward function. It would enhance clarity if the authors summarized the algorithm in a table format.
2. The authors primarily use the total number of gates in the circuit, including both single-qubit and two-qubit gates, as the main evaluation metric for circuit size. However, two-qubit gates are more challenging to implement in practical devices and are more susceptible to noise. It would be beneficial to separately compare the counts of single-qubit and two-qubit gates in the circuits generated by the proposed algorithm against those produced by existing algorithms.
3. Typos: Line 256: "a the target".

**Questions:**

The questions are included in Weaknesses.

---

> ### Author Response · Authors · 2024-11-27
> **Rebuttal by Authors**
>
> Thank you for the feedback and for appreciating our work. We hope that the answers below sufficiently address your concerns. Otherwise, we would be happy to engage further.
>
> > As someone who is not well-versed in RL algorithms ... if the authors summarised the algorithm in a table format.
>
> We regret this fallacy in the presentation. We have summarised the precise RL formulation used for QSP in table format. In addition, we have added more extensive outlines for specific subparts of the algorithm in the appendix. We hope that this makes the presentation more clear.
>
> > The authors primarily use the total number of gates in the circuit ... against those produced by existing algorithms
>
> Absolutely! That is a very important concern, and we have added additional experiments in Appendix D.1. to benchmark this. Please note that our agents are never explicitly trained to minimize two-qubit gates, and are trained with single- and two-qubit gates placed on an equal footing. Whereas, the Bravyi et al work explicitly optimizes the two-qubit gate count. However, as our experiments on entanglement dynamics showed, the agent’s actions do not display much redundancy. To confirm this, we also benchmarked our trained agents on the number of CNOTs (the single qubit gate benchmarks are thus inferred by comparing the CNOTs and total number of gates). Here are the results for CNOTs:
>
> | Algorithm (two-qubit gate count →) | 5-qubit | 6-qubit | 7-qubit | 9-qubit |
> |:-----------------------------------|:--------|:--------|:--------|:--------|
> | A-G | 9.12 ± 3.29 | 14.92 ± 3.85 | 21.34 ± 4.16 | 38.22 ± 5.46 |
> | B-M | 7.56 ± 2.46 | 11.89 ± 2.81 | **16.30 ± 2.86** | **26.46 ± 3.17** |
> | RL (linear connectivity) | 10.16 ± 4.16 | 14.50 ± 7.34 | 18.44 ± 4.10 | - |
> | RL (full connectivity) | **6.08 ± 2.45** | **9.13 ± 2.28** | 19.52 ± 7.71 | 34.20 ± 13.48 |
>
> Here, A-G refers to the classical algorithm of Aaronson and Gottesman, and B-M the optimized algorithm by Bravyi et al. Notice that we perform well with respect to CNOT gates, despite having given no bias towards minimizing the number of two-qubit gates. This further emphasizes our efficiency in zero-shot state preparation.
>
> [A] Aaronson et al, Improved simulation of stabilizer circuits, Phys. Rev. A 70, 052328 (Nov. 2004)
>
> [B] Bravyi et al, Clifford Circuit Optimization with Templates and Symbolic Pauli Gates, Quantum 5, 580 (2021)
>
> > Typos: Line 256: "a the target".
>
> Thank you, this is fixed!
>
> ---
> We are truly grateful for the effort and attention you have devoted to reviewing this work, as well as for the valuable feedback that has guided improvements in its quality and clarity. We hope our responses and the additional evidence provided have resolved your concerns, and we would be grateful if this could be reflected in a revised score. We are, of course, happy to engage further if you have additional queries or suggestions.

---

> > ### Comment · Reviewer_3SCv · 2024-12-03
> >
> > Thanks for the author's response. I have no further questions on my end. However, after reviewing the comments from the other reviewer, I agree with Reviewer GvX9’s concern that the title referencing 'zero-shot learning' may falsely suggest additional contributions that are not actually part of this work. If the authors can clarify this issue, I will maintain my current score."

---

> > > ### Author Response · Authors · 2024-12-03
> > > **Response by Authors**
> > >
> > > Thank you for your engagement - we're glad to hear your concerns have been resolved. Based on the feedback from reviewer GvX9 and others, we have now clearly explained in the paper what we mean by zero-state preparation (Lines 70-72 in the revised document), to prevent any confusion or misinterpretations about the contributions of this work. In particular, to quote the revised version, "*An agent that does not need re-training to prepare unseen
> > > states will be called zero-shot in this work, to emphasize successful generalization to states not seen
> > > during training*".
> > >
> > > We hope that resolves this issue, but we're also willing to edit the title slightly to "Zero-shot generalization: Quantum state preparation with immediate inference using RL". We also welcome any other suggestions you might have in this regard. We're grateful for your support of this work."

---

> ### Author Response · Authors · 2024-12-01
> **Discussion phase ending soon - awaiting your response**
>
> Dear Reviewer,
>
> Thank you for your encouraging feedback. We have worked hard to address your concerns by greatly improving the presentation of the method and benchmarking the count of the two-qubit gates separately for our method against existing baselines.
>
> Since the author-reviewer discussion period ends soon, we shall be grateful if you could consider upgrading your score to reflect the overall improvement due to these enhancements. We also welcome any further questions, concerns, or suggestions. Many thanks!
>
> Best regards,
>
> The authors

---

### Official Review · Reviewer_5w9Q · 2024-11-05

**Soundness:** 1
**Presentation:** 1
**Contribution:** 2
**Rating:** 3
**Confidence:** 5

**Summary:**

In this paper, the authors employ reinforcement learning (RL) to the task of quantum state preparation (QSP) on physically relevant states. Additionally, their method enables zero-shot preparation for any target state when the system size is fixed.
Numerical results for systems with up to $9$ qubits show that their method generates circuits up to $60$% shorter than the baseline methods evaluated in the study.

**Strengths:**

The paper does a reasonable job introducing the quantum state preparation task and, from the presented difficulties of the domain, manages to derive its adaptations to standard RL employed to tackle this domain. The adaptations presented are sound and straight-forward. The results give a nice guideline for future research in that direction.

**Weaknesses:**

- The presentation of the paper is somewhat convoluted and challenging to follow; clearer descriptions would be helpful.
  - The contributions could be better highlighted, such as the introduction of the moving goalpost reward (MGR) for the QSP task.
  - The methodology and results, presented through plots, are introduced quite late in the paper. Providing clearer scenario descriptions of the experiments would improve readability.
  - A clearer contrast with related work would be helpful.
  - Figure legends could be enlarged for improved readability.
  - To aid understanding of the MGR function, a figure illustrating its workings would be valuable.

- The paper appears to misinterpret the QSP definition by assuming access to the target state $|\psi\rangle$. Typically, in QSP, one has access only to a succinct (classical) description of the target state, not the circuit that implements it. However, in this approach, the authors start each episode with $|\psi\rangle$ as the initial state and then find a circuit to prepare $|\mathbf{0}\rangle^{\otimes n}$ for zero-shot state preparation. This approach may not be physically feasible in a lab; if the circuit for $|\psi\rangle$ is known, the rationale for using RL (or any method) to find a circuit that prepares $|\mathbf{0}\rangle^{\otimes n}$ and then inverts it is unclear. It might be argued that the given $|\psi\rangle$ circuit is too deep or not NISQ-friendly, necessitating a more compact architecture, though this would still be costly in a real lab setting.

- Insufficient comparison with relevant state-of-the-art methods for the quantum state preparation task:
  - RL methods: [1-4]
  - ML methods: [5,6]
  - SAT-based methods: [7,8]

- The algorithm's performance under common noise models, such as state preparation and measurement (SPAM), bit flip, phase flip, depolarizing noise, and thermal relaxation is not evaluated. These noise sources are standard in quantum computing.

- Several claims in the paper lack theoretical or numerical support. Ablation studies providing numerical evidence would help support these claims:
  - Line 306-308: "However, the cumulative reward obtained ..... terminating the episode."
  - Line 803-806: "Finally, for the linear-connectivity agents, .... faster with this term."

- The paper lacks open-source code and a reproducibility statement.


[1] Fosel, Thomas, et al. "Quantum circuit optimization with deep reinforcement learning", arXiv:2103.07585 (2021)\
[2] Patel, Yash J., et al. "Curriculum reinforcement learning for quantum architecture search under hardware errors",  ICLR (2024)\
[3] Zen, Remmy, et al. "Quantum Circuit Discovery for Fault-Tolerant Logical State Preparation with Reinforcement Learning", arXiv:2402.17761 (2024)\
[4] Kremer, David, et al. "Practical and efficient quantum circuit synthesis and transpiling with Reinforcement
Learning", arXiv:2405.13196 (2024)\
[5] Wang, Hanrui, et al. "Quantumnas: Noise-adaptive search for robust quantum circuits." 2022 IEEE International Symposium on High-Performance Computer Architecture (HPCA)\
[6] Wu, Wenjie, et al. "Quantumdarts: differentiable quantum architecture search for variational quantum algorithms", ICML (2023)\
[7] Peham, Tom, et al. "Depth-Optimal Synthesis of Clifford Circuits with SAT Solvers", QCE (2023)\
[8] Schneider, Sarah, et al. "A SAT Encoding for Optimal Clifford Circuit Synthesis", ASPDAC (2023)

**Questions:**

- see above.

- HSH gate in the gateset:
  - What is the rationale behind including HSH gate in the gateset $G$ (action space)? Because in principle the agent should learn the symmetry on $S$ w.r.t. to $X$ and $Z$ operations by itself. I believe this inductive bias is introduced during the learning process while the whole point of using RL is not to have any kind of bias.

  - Also HSH gate has three single qubit gates. In Table 1, do you count it as one gate for all the methods? Do you use the same gateset for all the methods for a fair comparison?

  - Can you provide an ablation study with and without HSH gate to see how the number of gates differ between these two settings for your method?

- Why is there no circuit size numbers for $9$-qubit DRL (local gates) in Table 1?

- In Fig. 3(d), $n=9$, can you please explain the behaviour of the circuit size for depth $t=1, 2$ signficantly larger for your method compared to other two baselines?

- How does the algorithm perform for the QSP task when the target states which are complex and physically relevant but are not classically simulable?

---

> ### Author Response · Authors · 2024-11-27
> **Rebuttal by Authors (1/4)**
>
> Many thanks for your thoughtful, constructive, and insightful review. We are immensely grateful to you for thoroughly reading our paper and raising important questions. We hope that the answers below sufficiently address your concerns.
>
> > The presentation of the paper is somewhat convoluted and challenging to follow; clearer descriptions would be helpful.
>
> Thank you very for bringing this to our attention, and for the great suggestions. We have re-written and re-organized parts of the paper to make it clearer and more complete, based on your feedback. We clearly specify our contributions in the introduction. We have re-written parts of the experiments section to be more elaborate and self-contained. All experiment scenarios and assumptions have been motivated and remarked on at the start of the section. We also improve our contrast with previous work and include some new works in the related-work section. We have improved the quality of the figures and added a new figure (Fig. 2) benchmarking the MGR reward and other rewards discussed in the revised paper, to further justify our choice of reward.
>
> We hope that the presentation issues raised have been satisfactorily addressed.
>
> > The paper appears to misinterpret the QSP definition by assuming access to the target state $\ket{\psi}$. Typically, in QSP, one has access only to a succinct (classical) description of the target state, not the circuit that implements it. However, in this approach, the authors start each episode with $\ket{\psi}$ as the initial state and then find a circuit to prepare $\ket{0}^{\otimes n}$ for zero-shot state preparation. This approach may not be physically feasible in a lab; if the circuit for $\ket{\psi}$ is known, the rationale for using RL (or any method) to find a circuit that prepares $\ket{0}^{\otimes n}$ and then inverts it is unclear. It might be argued that the given $\ket{\psi}$ circuit is too deep or not NISQ-friendly, necessitating a more compact architecture, though this would still be costly in a real lab setting.
>
> We are thankful for this comment and apologize for a potential fallacy in the presentation of the state preparation problem. In particular, we do not need access to a quantum system already prepared in the state $\ket{\psi}$ to run the algorithm. We reaffirm working in the QSP setting, wherein one clear task that our algorithm solves is: given (the classical description of) a state to be prepared in a lab, provide an optimal circuit for it. Furthermore, the access model is amenable to modifications as we clarify in the future work section of the revised manuscript as well as the general response.
>
> > Insufficient comparison with relevant state-of-the-art methods for the quantum state preparation task.
>
> Thank you for the detailed suggestions. We shall incorporate comparisons with these into our paper and performance results. Below is a comparison between our work and the references mentioned.
>
> > [1] Fösel, Thomas, et al. "Quantum circuit optimization with deep reinforcement learning", arXiv:2103.07585 (2021)
>
> While this is a useful paper, it addresses the related problem of quantum circuit optimization.
>
> > [2] Patel, Yash J., et al. "Curriculum reinforcement learning for quantum architecture search under hardware errors", ICLR (2024)
>
> Their CRLQAS algorithm addresses the case of circuits containing parameterized rotation gates, which do not yield stabilizer states; at this point, it would not be possible to compare performance between the two.
>
> > [3] Zen, Remmy, et al. "Quantum Circuit Discovery for Fault-Tolerant Logical State Preparation with Reinforcement Learning", arXiv:2402.17761 (2024)
>
> Thank you for this suggestion. It is interesting to note that the circuits prepared by our agent for these states, although never trained to prepare them, are only $3$-$4$ gates longer than the circuits described in [3], despite agents being re-trained for each circuit in [3].
>
> > [4] Kremer, David, et al. "Practical and efficient quantum circuit synthesis and transpiling with Reinforcement Learning", arXiv:2405.13196 (2024)
>
> Thank you for this great reference. Indeed, it attacks the Clifford problem as well with a reverse-preparation approach and managed to scale up to eleven qubits. We include it in our manuscript as related work and contrast it from our approach. Our approach is much more sample-efficient; we require $10$-$20$M samples (equal to number of training steps) to train the agent on $7$ qubits, while the model in the referenced work uses $1$B+. We believe that this is because of our improved reward function: in particular, the key driver for the scalability and sample-efficiency is not the zero-shot preparation, but rather the reward function.

---

> ### Author Response · Authors · 2024-11-27
> **Rebuttal by Authors (2/4)**
>
> > [5] Wang, Hanrui, et al. "Quantumnas: Noise-adaptive search for robust quantum circuits." 2022 IEEE International Symposium on High-Performance Computer Architecture (HPCA)
>
> According to a preliminary reading, this paper seeks to search for noise-resilient variational quantum circuits, which is a different goal from ours.
>
> > [6] Wu, Wenjie, et al. "Quantumdarts: differentiable quantum architecture search for variational quantum algorithms", ICML (2023)
>
> This paper, like [2], seems to attack the architecture search problem for VQCs.
>
> > [7] Peham, Tom, et al. "Depth-Optimal Synthesis of Clifford Circuits with SAT Solvers", QCE (2023)
> > [8] Schneider, Sarah, et al. "A SAT Encoding for Optimal Clifford Circuit Synthesis", ASPDAC (2023)
>
> These are valuable methods that can prepare Clifford circuits optimally, and we shall not miss citing them in related work on Clifford synthesis. With respect to performance comparison, we believe there is a fundamental difference in the problem being solved, i.e. state preparation (our work) vs circuit preparation [7, 8]. An agent preparing a state can optimize over all circuits that send the fiduciary to the state; circuit preparation can only optimize over different decompositions of a fixed unitary, and so will naturally require more gates. In light of this, we do not think it would be fair to benchmark a state preparation agent with one that prepares circuits. Having said this, a future work is to extend our ideas to Clifford circuit preparation, using the fact that the pair of stabilizers and destabilizers specifies a Clifford unitary uniquely. These references will be much more valuable in this case.
>
> > The algorithm's performance under common noise models, such as state preparation and measurement (SPAM), bit flip, phase flip, depolarizing noise, and thermal relaxation is not evaluated. These noise sources are standard in quantum computing.
>
> As rightly pointed out, we have not accounted for robustness of the generated circuits to noise. We had anticipated this issue during the course of our work, and making reverse-preparation work to prepare fault-tolerant circuits is a future work. However, we tested the agent's robustness to the equivalent noise in the tableau representation. The agent is able to quickly correct it using a number of gates close to the number of bit/phase flips itself. This is due to the agent's understanding of the entanglement correlations (Sec. 5, Entanglement Dynamics), offering monotonic improvement towards the state $\ket{0}^{\otimes n}$ despite errors.
>
> > Several claims in the paper lack theoretical or numerical support. Ablation studies providing numerical evidence would help support these claims:
> > Line 306-308: "However, the cumulative reward obtained ..... terminating the episode."
> > Line 803-806: "Finally, for the linear-connectivity agents, .... faster with this term."
>
> Thank you for pointing these out; we are grateful to see a thorough reading of our work.
>
> For the claim on Lines 306-308, the idea is that it is simple for the agent to choose a gate that does not increase fidelity. Indeed, in our revised paper (Sec 4.2), we show that *an overwhelming majority* of gates lead to the same fidelity. This is partially because of the limited distinct values -- only inverse powers of 2 -- the fidelity can take on. With the reward being the fidelity to the target, it is optimal (maximum return) for an RL agent to prepare a state with fidelity $0.5$ to the target, followed by staying at fidelity $0.5$ till forced truncation of the episode --- this is not what we wish. Hence the comment that this reward does not reflect our true goal. Thank you for pointing this out, we have updated our manuscript with a detailed explanation. Further, we provide numerical evidence for the same, by training an agent on this reward and indeed noticing that it keeps the fidelity at $0.5$ at convergence. Please see Fig. 2 in the revised manuscript.
>
> For the second claim on Lines 803-806, we found that the Jaccard distance helps guide our agent faster, especially at large $n$. For example, at 6 qubits, our experiments show that without Jaccard distance, we take 20K episodes to reach an average fidelity close to $1$, but only 15K with this term.
>
> > The paper lacks open-source code and a reproducibility statement.
>
> Thank you for your concern. We take open-source research seriously, and had indeed put up our code with running instructions in the attached supplementary information with our original submission.

---

> ### Author Response · Authors · 2024-11-27
> **Rebuttal by Authors (3/4)**
>
> > HSH gate in the gateset: What is the rationale behind including HSH ... any kind of bias.
>
> Thank you for the opportunity to clarify this point. We make use of this gate to make the action space symmetric, as illustrated in the paper. This symmetry is useful since the tableau is also symmetric in $X$ and $Z$. As you correctly inferred, $HSH$ is an inductive bias w.r.t the original gate-set of $H$, $S$ and CNOT. However, we would like to point out the fact that $S$ is a $Z(\pi/2)$ gate, and $HSH$ is a $X(\pi/2)$ gate. In many experiments demonstrating state of the art single and two qubit gate fidelities, there is no difference to the cost of doing $\pi/2$ rotations around the $X$ or $Z$ axes (for instance see [A] for a neutral atom computer).
>
> [A] Evered et al. High-fidelity parallel entangling gates on a neutral-atom quantum computer. Nature 622, 268–272 (2023)
>
> > Also $HSH$ gate has three single qubit gates. In Table 1, do you count it as one gate for all the methods? Do you use the same gateset for all the methods for a fair comparison?
>
> As mentioned in the response to the above comment, we do count it as one gate, since it has the same costs of implementation as $S$ on many promising quantum hardware. Actually, except the $HSH$ gate, the baselines [A, B] have significantly more power, being allowed the gateset $\{H, S, \text{CNOT}, \text{SWAP}, X, Y, Z\}$.
>
> [A] Aaronson et al, Improved simulation of stabilizer circuits, Phys. Rev. A 70, 052328 (Nov. 2004)
>
> [B] Bravyi et al, Clifford Circuit Optimization with Templates and Symbolic Pauli Gates, Quantum 5, 580 (2021)
>
> > Can you provide an ablation study with and without HSH gate to see how the number of gates differ between these two settings for your method?
>
> Thank you for the question. Here are the results of training an agent for $6$-qubit stabilizer states, run for full connectivity with gate-sets {H, S, HSH, CNOT} and {H, S, CNOT} respectively.
>
> | Metric      | With HSH               | Without HSH           |
> |-------------|------------------------|-----------------------|
> | Gate Count  | 17.86 ± 2.88           | 22.80 ± 3.88          |
> | CNOT Count  | 9.13 ± 2.28            | 9.90 ± 2.10           |
>
> > Why is there no circuit size numbers for $9$-qubit DRL (local gates) in Table 1?
>
> We were unable to train this agent to successful convergence, despite our many tries. We still believe that it can be done with a suitable architecture and some optimization, however, we have not been successful so far. Please note that the local agents are harder to prepare compared to full connectivity agents, in part due to local connectivity requiring substantially longer circuits to implement the same unitary. Actually, they appear to be the connectivity with the longest average circuit size (Tab. 3 in [A]). It is an important further work for us to scale our algorithm further.
>
> [A] Kremer et al. "Practical and efficient quantum circuit synthesis and transpiling with Reinforcement Learning", arXiv:2405.13196 (2024)
>
> > In Fig. 3(d), $n = 9$, can you please explain the behaviour of the circuit size for depth $t = 1,2$ significantly larger for your method compared to other two baselines?
>
> Thank you for this question. The increased gate counts for $t = 1, 2$ are due to multiple pairs of adjacent identical CNOT gates in the produced circuit. We describe how these pairs come about. Consider a brickwork state $\ket{\psi}$ with depth $t \leq 2$. Since our agent has full connectivity, its first guess is quite likely a CNOT gate between non-adjacent qubits. However, $t = 1$ cannot capture such an entanglement. What we mean by this is that any circuit preparing $\ket{0}$ from $\ket{\psi}$ cannot have an entangling gate between qubits at a distance more than two from each other that \emph{does not} vanish upon circuit optimization -- otherwise, one can argue that a bipartition of the qubits is entangled, but $t \leq 2$ forces it to be un-entangled.
>
> Luckily, the agent is able to quickly correct its incorrect guess by repeating it. This process may repeat multiple times, till the agent finds the right local gates to prepare the state. Essentially, any algorithm that *guesses* the first few moves will struggle at low brickwork depth, since such states support very limited entanglement configurations with most guesses infeasible for a circuit preparing the state. However, for $t \geq 3$, entanglement between farther qubits is feasible, and the agent does not need to immediately correct a guess that entangled non-adjacent qubits but can use it to move forward in the preparation. This explains more generally the quickly decaying circuit size as one moves from $t = 1$ to $t = 5$, after which any two qubits may be entangled, allowing the agent to keep its initial guesses and move forward.
>
> We thank the reviewer once again for this insightful question that reflects an important facet of the agent's entanglement dynamics.

---

> ### Author Response · Authors · 2024-11-27
> **Rebuttal by Authors (4/4)**
>
> > How does the algorithm perform for the QSP task when the target states which are complex and physically relevant but are not classically simulable?
>
> Our agents are currently restricted to preparing stabilizer states only, since our architecture takes a stabilizer tableau as input. However, we have already used the same framework and reward function to prepare agents that successfully solve the zero-shot general state preparation problem for up to three qubits with $\varepsilon = 0.95$.
>
> However, in this work, our focus is on stabilizer states, and the general setting requires more comprehensive investigation with suitable adaptations, which is beyond the current scope.
>
> ---
>
> We sincerely appreciate your time, effort, and the excellent suggestions that have greatly enhanced the clarity and quality of this work. We trust that our clarifications and additional evidence have addressed your concerns, and we would be grateful if this could be reflected in a revised score. We remain committed to further discussion should you have additional questions or suggestions.

---

> ### Author Response · Authors · 2024-12-01
> **Discussion phase ending soon - awaiting your response**
>
> Dear Reviewer,
>
> Thank you for your thoughtful feedback. We have worked hard to address your concerns by greatly improving the presentation, contrasting each work provided, clarifying the QSP definition and adding missing evidence to support our claims. We also strived to address the issue of gate-set, anomalous behavior of the 9-qubit agent at low depth and the generalization of our framework beyond stabilizer states.
>
> Since the author-reviewer discussion period ends soon, we shall be grateful if you could consider upgrading your score to reflect the overall improvement due to these enhancements. We also welcome any further questions, concerns, or suggestions. Many thanks!
>
> Best regards,
>
> The authors

---

### Official Review · Reviewer_PcnZ · 2024-11-07

**Soundness:** 2
**Presentation:** 2
**Contribution:** 2
**Rating:** 5
**Confidence:** 5

**Summary:**

The paper proposes a zero-shot quantum state preparation approach. RL has been used.
A moving goal post reward function has been designed. Training is performed on less than 0.0001% of the
state space and then generalized.

**Strengths:**

1. Upto 60% shorter circuit depths are obtained.
2. Generalized to more than 2 qubits without re-training.

**Weaknesses:**

1. Proof of Prop. 2 of convergence is not completely convincing. Conditions
under which it is valid need rigorous statement.
2. No guarantee that the potential function will be strictly convex. No consideration
is given to this.
3. It is not clear why retraining is not required. How general is this is not stated.

**Questions:**

1. Why retraining is not required needs further explanation and study?
2. Why was the 9 qubit agent not trained till convergence?

---

> ### Author Response · Authors · 2024-11-27
> **Rebuttal by Authors**
>
> Thank you for your feedback. We hope that the answers below sufficiently address your concerns.
>
> > Proof of Prop. 2 of convergence is not completely convincing. Conditions under which it is valid need rigorous statement.
>
> Proposition 2 (renamed Prop. 4 in the revised paper and proved in Appendix B.1) holds for any real function $\Phi: \mathcal S\to \mathbb R$, similar to Theorem 1 in [A]. The key idea involved is a telescoping sum that leaves only the initial and maximum terms as the cumulative reward. We have elaborated on the potential function and re-written the statement of the proposition to make it clearer. Thank you for pointing this out.
>
> > No guarantee that the potential function will be strictly convex. No consideration is given to this.
>
> As we have mentioned above, the proposition holds for any function $\Phi: \mathcal S \to\mathbb R$. We apologize for the usage of the term ''potential function'' for $\Phi$, which obviously lends itself naturally to questions of convexity. We have explicitly defined potential functions in the revised manuscript to avoid ambiguity.
>
> > It is not clear why retraining is not required. Why retraining is not required needs further explanation and study?
>
> By varying the start state in each episode, we are implicitly training the agent to be able to prepare $\ket{0}^{\otimes n}$ from every $\ket{\psi}$. Since inverting the circuit yields a circuit preparing $\ket{\psi}$ from $\ket{0}^{\otimes n}$, we are indirectly forcing the agent to prepare any given state $\ket{\psi}$. Thus, one training phase of the agent solves (in theory) the problem of preparing every state $\ket{\psi}$, and so retraining is not needed. Experiments and our theoretical bound show that one training phase is also sufficient in practice: the agent is then guaranteed to generalize to at least $95$% of the state space. Thank you for this point; we have made the exposition clearer and more formal in the revised manuscript, under the Methods section.
>
> > How general is this is not stated.
>
> The formulation as well as our reward function actually applies in full to the general (approximate) quantum state preparation problem, please see the Methods section for more details. However, this work is fully devoted to the important albeit limited case of stabilizer state preparation. We find that the insights drawn from the agent dynamics and agent generalization indeed does hint toward the methodology being applicable to non-stabilizer states. Moreover, with suitable modifications of the action sets, we have been successful in training a zero-shot agent for the general state preparation problem on $3$ qubits, and are currently working on generalizing to non-stabilizer states. We have clarified more on this issue in the general response.
>
> > Why was the $9$ qubit agent not trained till convergence?
>
> We apologize for the confusion, and would like to point out that we had actually trained it to convergence. Given that the average return at the time was (slightly) negative, we believed the solution was currently sub-optimal and that the reward would eventually rise, leading to the comment in the paper. However, later analysis showed that the negative reward is expected, since we had chosen $\gamma = 0.9$ for this experiment and the average episode length was $T > 40$, the constant term $-\alpha$ outweighed benefits. In particular, the sum of the positive $\gamma^T\Phi^*$ and $\gamma^j\max\{\Phi(s_i)\,:\,1\leq i\leq j\}$ terms were outweighed by the $-\alpha$ term, i.e. $-\alpha\sum_{i=0}^{T-1}\gamma^i$, leading to net (slightly) negative reward. This was not the case with the other experiments where $\gamma = 0.99$, which yielded positive reward upon successful convergence. So we concluded, forgetting that $\gamma$ was $0.9$ for the $9$-qubit case, that the agent was not done and would later perform better, though an increase in training time did not yield a substantial improvement -- which is obvious.
>
> **TL;DR** we simply did not recognize that our agent had actually successfully converged, and converged to a good policy. In fact, we show in the revised manuscript that the $9$-qubit agent generalizes to at least $95$\% of the $9$-qubit stabilizer state space -- a massive generalization. We apologize again for the confusion.
>
> [A] Ng, A. et al. “Policy Invariance Under Reward Transformations: Theory and Application to Reward Shaping.” International Conference on Machine Learning (1999).
>
> ---------------------------------------
> Many thanks for the thoughtful and constructive suggestions that have significantly contributed to the quality of this work. We believe our clarifications and the additional evidence address the issues raised, and we would be grateful if this could be reflected in an updated score. Should there be further questions or comments, we remain eager to engage in further discussion.

---

> ### Author Response · Authors · 2024-12-01
> **Discussion phase ending soon - awaiting your response**
>
> Dear Reviewer,
>
> Thank you for your thoughtful feedback. We have made our best attempt to address your concerns in our response, by revising the proofs and stating conditions, clarifying and improving the exposition on why retraining is not required. We have also elaborated on the broader applicability of our method and explained the issue of 9-qubit agent convergence.
>
> Since the author-reviewer discussion period ends soon, we shall be grateful if you could consider upgrading your score to reflect the overall improvement due to these enhancements. We also welcome any further questions, concerns, or suggestions. Many thanks!
>
> Best regards,
>
> The authors

---

> > ### Comment · Reviewer_PcnZ · 2024-12-03
> > **Comment**
> >
> > Thanks to the authors for the clarifications and the responses. I have no further questions but
> > based on discussions and an overall assessment, I decide to keep my score unchanged.

---

> > > ### Author Response · Authors · 2024-12-03
> > > **Response by Authors**
> > >
> > > Dear Reviewer,
> > >
> > > Thank you for your feedback. We greatly value your input and have worked diligently to address your concerns. Specifically, we have:
> > >
> > > - Revised the proofs and explicitly stated the required conditions.
> > > - Clarified and improved the exposition regarding why retraining is not necessary.
> > > - Expanded on the broader applicability of our method.
> > > - Provided a detailed explanation of the 9-qubit agent convergence issue.
> > >
> > > Additionally, other reviewers have acknowledged that their concerns have been adequately addressed.
> > >
> > > We kindly request you to revisit your score in light of these improvements. If there are any remaining questions or additional points of concern that we could address to further improve the paper and earn a higher evaluation, we would be sincerely grateful to hear them.
> > >
> > > Thank you once again for your time and valuable feedback!
> > >
> > > Best regards,
> > >
> > > The authors

---

### Author Response · Authors · 2024-11-27
**General Response by Authors (1/2)**

We are grateful to the reviewers for their time and insightful comments; and to the area, senior area, and program chairs for their service.
We are glad to note that the reviewers appreciated several aspects of this work including **efficiency** (PcnZ, 3SCv, hnhQ, GvX9), the novel **reward** function (3SCv, GvX9), **theoretical analysis** (3SCv, GvX9, crcX) and **relevance** to the quantum community (hnhQ, GvX9).

We have tried our best to address all the questions and concerns raised by the reviewers including the minor ones. We summarize below how we have addressed the key concerns here in addition to the point-by-point reviews to the reviewers.

1. **Stabilizer and non-stabilizer states**. This comment is in regards to some referees' concerns about the general applicability of the framework and the reason for implementation on Clifford circuits. We apologize for the possible lack of clear explanation in the manuscript, and clarify this point in detail here as well as in the paper. We would like to stress the fact that the classical simulability of stabilizer states does not imply triviality of state preparation. Moreover, there is a plethora of new physics in quantum dynamics that has seen light in the recent years owing to the classical simulability of Clifford circuits, discoveries which would not have been possible without this numerical convenience (for instance see [A-D]). Yet, many of those aspects generalize to non-Clifford dynamics and circuits. The takeaway is that by leveraging the classical simulation possibility of a class of quantum states, which form a 3-design and capture many properties of universal quantum dynamics, we are able to gain enhanced insight into the possibilities of leveraging reinforcement learning methods for quantum state preparation. Indeed, the story is not complete and this method needs to be studied in the full generality of the Hilbert space. Thus, looking forward, we provide important future directions to achieve this generality while concluding the paper.

[A] Li, Yaodong, Xiao Chen, and Matthew PA Fisher. "Quantum Zeno effect and the many-body entanglement transition." Physical Review B 98.20 (2018): 205136.

[B] Li, Yaodong, Xiao Chen, and Matthew PA Fisher. "Measurement-driven entanglement transition in hybrid quantum circuits." Physical Review B 100.13 (2019): 134306.

[C] "Measurement-induced entanglement and teleportation on a noisy quantum processor." Nature 622, no. 7983 (2023): 481-486.

[D] Bao, Yimu, Maxwell Block, and Ehud Altman. "Finite-time teleportation phase transition in random quantum circuits." Physical Review Letters 132.3 (2024): 030401.

2. **The QSP setting**. In response to comments regarding the information the agent has access to, we would like to clarify that we are indeed working in the typical QSP setting, but with relaxed conditions for this work where we are allowed the full (classical) stabilizer tableau of the state to the algorithm. This is already relevant when one wants to implement a circuit conditioned on connectivity and gate-sets to prepare a state one knows about in advance in the lab. The algorithm provides the circuit that the experimentalist would implement. Further, we would like to point out that the inversion procedure is only for training the algorithm; after the fact, we only need access to the target state to find a circuit to generate it, reaffirming the QSP paradigm. Moreover, while one would ideally like the query model to be, say, local Pauli measurements on the target state, we have a relaxed query model wherein the full tableau is available as a query. This work indeed establishes the fact that training very general agents using reinforcement learning is possible. An immediate future work that we are pursuing is changing the query model and hence bringing it closer the setting of preparing states given succinct classical descriptions.

3. **Zero-shot terminology**. We agree that usage of "zero-shot" without some additional context could be confusing. Indeed, multiple zero-shot settings have received attention in the community such as predicting images from a class without having been presented any examples of that class during training, or more recently, in the context of so called 'zero-shot prompting' in LLMs.  Here, we use the term "zero-shot" to convey the successful preparation of a state without ever having access to it. We have clarified our usage of this definition of zero-shot in the introduction of the revised manuscript. To quote, "*An agent that does not need re-training to prepare unseen
states will be called zero-shot in this work, to emphasize successful generalization to states not seen during training.*"

---

> ### Author Response · Authors · 2024-11-27
> **General Response by Authors (2/2)**
>
> 4. **Convergence of the nine-qubit experiment**. This comment is in relation to a common concern regarding the incompleteness of the training in the nine-qubit experiment. We apologize for the fallacy in presentation, and would like to point out that it is indeed trained to convergence as we clarify here. Given that the average return at the time was small and negative, we believed the solution was currently sub-optimal and that the reward would eventually rise, leading to the comment in the paper. However, further analysis showed that the negative reward is expected, since we had chosen $\gamma = 0.9$ for this experiment and the average episode length was $T > 40$, the constant term $-\alpha$ outweighed benefits. In particular, the sum of the positive $\gamma^T\Phi^*$ and $\gamma^j\max\{\Phi(s_i)\,:\,1\leq i\leq j\}$ terms were outweighed by the $-\alpha$ term, i.e. $-\alpha\sum_{i=0}^{T-1}\gamma^i$. This was not the case with the other experiments where $\gamma = 0.99$, which yielded positive reward upon convergence. Nevertheless, the $9$-qubit agent had converged, and converged to a good policy.
>
> TL;DR we simply did not recognize that our agent had actually successfully converged to a good policy.
>
> In fact, we have now been able to obtain a much tighter bound for the generalization success, which confirms that with probability at least $1-10^{-10} \approx 1$, the $9$-qubit agent generalizes to at least 95\% of the $4.3\times 10^{16}$-sized state space. This is explained and proven using a concentration bound in the revised manuscript.
>
> ----------
> We thank reviewers again for their very constructive comments.

---

### Meta-Review · Area_Chair_d38n · 2024-12-19

**Metareview:**

The authors introduce a deep reinforcement learning (RL) framework for quantum state preparation (QSP), specifically targeting stabilizer states. By employing a novel moving-goalpost reward (MGR) function and a reverse-generation approach, the method demonstrates zero-shot generalization, enabling the preparation of unseen stabilizer states without retraining. Simulations conducted on systems with up to nine qubits highlight the effectiveness of the proposed method.

While the integration of reinforcement learning with stabilizer state preparation is noteworthy, the submission is hindered by several critical issues:
1. The study mostly focuses on stabilizer states, limiting its broader applicability.
2. The theoretical guarantees for generalization success are considered loose and lack rigorous validation.
3. The absence of evaluations under realistic quantum noise models, such as depolarizing or SPAM noise, undermines the practical relevance.
4. The presentation was critiqued for devoting excessive space to background material while omitting critical details of the RL implementation and methodology.

**Additional Comments On Reviewer Discussion:**

After the discussion, the authors and reviewers resolved several unclear issues and points of confusion, including acknowledging the limitation to stabilizer states, appending preliminary results for non-stabilizer states, and providing new bounds on generalization success rates. Despite these efforts, the authors were unable to fully convince the reviewers of the broader impact and key contributions of the proposed methods.

---

### Decision · Program_Chairs · 2025-01-22

Reject